# Fault Diagnosis of Rotating Machinery: A Highly Efficient and Lightweight Framework Based on a Temporal Convolutional Network and Broad Learning System

**DOI:** 10.3390/s23125642

**Published:** 2023-06-16

**Authors:** Hao Wei, Qinghua Zhang, Yu Gu

**Affiliations:** 1College of Information Science and Technology, Beijing University of Chemical Technology, Beijing 100029, China; 2School of Automation, Guangdong University of Petrochemical Technology, Maoming 525000, China; 3Beijing Advanced Innovation Center for Soft Matter Science and Engineering, Beijing University of Chemical Technology, Beijing 100029, China

**Keywords:** broad learning system, fault diagnosis, temporal convolutional network, rotating machinery

## Abstract

Efficient fault diagnosis of rotating machinery is essential for the safe operation of equipment in the manufacturing industry. In this study, a robust and lightweight framework consisting of two lightweight temporal convolutional network (LTCN) backbones and a broad learning system with incremental learning (IBLS) classifier called LTCN-IBLS is proposed for the fault diagnosis of rotating machinery. The two LTCN backbones extract the fault’s time–frequency and temporal features with strict time constraints. The features are fused to obtain more comprehensive and advanced fault information and input into the IBLS classifier. The IBLS classifier is employed to identify the faults and exhibits a strong nonlinear mapping ability. The contributions of the framework’s components are analyzed by ablation experiments. The framework’s performance is verified by comparing it with other state-of-the-art models using four evaluation metrics (accuracy, macro-recall (MR), macro-precision (MP), and macro-F1 score (MF)) and the number of trainable parameters on three datasets. Gaussian white noise is introduced into the datasets to evaluate the robustness of the LTCN-IBLS. The results show that our framework provides the highest mean values of the evaluation metrics (accuracy ≥ 0.9158, MP ≥ 0.9235, MR ≥ 0.9158, and MF ≥ 0.9148) and the lowest number of trainable parameters (≤0.0165 Mage), indicating its high effectiveness and strong robustness for fault diagnosis.

## 1. Introduction

Rotating machinery, which consists of many rolling components, is the most widely used mechanical equipment [1] and is essential in many fields, such as aviation, transportation, and chemical industries. Due to technological advances in the manufacturing industry, rotating machinery has become increasingly complex and automated, increasing the requirements for the safe operation of rotating machinery [2]. However, some key components, such as bearings and gears, are susceptible to damage in complex and harsh operating environments, resulting in significant economic loss and human casualties [3,4]. Therefore, accurate and efficient fault diagnosis of the key components is crucial to guarantee the stable and safe operation of rotating machinery.

The development of artificial intelligence technology has attracted widespread attention for use in fault diagnosis based on monitoring data collected from sensors. Traditional intelligent fault diagnosis consists of two steps: (1) signal analysis of the monitoring data and manual features selection; (2) artificial fault assessment based on the signal analysis results and/or classification algorithms [2,5]. However, the effectiveness of the traditional fault diagnosis method relies largely on the experience of the maintenance personnel, leading to variable diagnosis results [6]. Besides, the non-stationarity of vibration signals and the nonlinear characteristics of rotating machinery in different scenarios can interfere with the extraction and identification of fault features [7]. Thus, it is necessary to research more intelligent methods to understand the relationship between the monitoring data and the equipment conditions to reduce the dependence of fault diagnosis on expert experience.

Deep learning-based fault diagnosis (DLFD), which can automatically extract features and establish the relationship between monitoring data and fault modes [8], has been increasingly applied to fault diagnosis research since it reduces the reliance on expert knowledge. DLFD methods can be categorized as: (1) fault diagnosis based on raw monitoring data and (2) fault diagnosis based on time–frequency domain-transformed monitoring data. Xie et al. [9] developed a deep one-dimensional (1-D) convolutional neural network (CNN) model to diagnose bearing faults using raw monitoring data. The effectiveness of the model was verified on the Case Western Reserve University (CWRU) dataset, achieving an average accuracy of 0.9958. Zhang et al. [10] proposed a deep residual network (ResNet) model based on residual learning to diagnose rotating machinery faults. The model used the raw data after 1-D convolution and achieved the highest accuracy of 0.9999 on the CWRU dataset. Although some studies, e.g., [9,10], successfully proposed methods to diagnose faults using raw data, the structure and complexity of the equipment produce non-stationary raw data [11], resulting in an inability to identify the fault features of the rotating machinery accurately in many applications. Appropriate methods to transform the data into the time–frequency domain can substantially increase the information extracted from the non-stationary raw signal [12]. Liang et al. [13] used time–frequency features extracted via wavelet transform (WT) and designed an effective fault diagnosis method consisting of a generative adversarial network and a CNN. The method achieved average accuracies of 0.9924 and 0.9789 on the CWRU dataset and a laboratory dataset, respectively. Zhao et al. [14] proposed a fault diagnosis model called multiple wavelet regularized deep ResNet. The model prevented overfitting for insufficient training data and improved the average accuracy by 8.19% and 3.62% compared with traditional CNNs and deep ResNets, respectively, on an experimental dataset. Some studies [13,14] designed outstanding fault diagnosis methods using the time–frequency features of the raw data. Nevertheless, some studies [15,16] have shown that the time–frequency analysis of the raw data could corrupt the structure of the vibration signals and alter the fault features, causing a one-sided fault feature representation and reducing the accuracy of fault identification. Therefore, it is necessary to research better methods to extract the complementary fault features and analyze the measured signals comprehensively to compensate for the shortcomings of existing fault diagnosis methods.

Another concern in fault diagnosis is the complexity (number of trainable parameters) of models. Many trainable parameters require high computing power and reduce the practicality of the models. The emergence of broad learning systems (BLSs) [17] has enabled the reduction in the computing cost for fault diagnosis while ensuring high accuracy. A BLS can be rapidly expanded by adding additional nodes through incremental learning, reducing the computational costs because training is not performed on the entire model [18]. BLSs have been used for fault diagnosis in the last five years. Fu et al. [18] designed an efficient fault diagnosis algorithm called adaptive BLS (ABLS) with two adaptive strategies to accelerate algorithm convergence and prevent under-fitting and over-fitting. Zhao et al. [19] proposed a fault diagnosis framework for rotors based on principal component analysis (PCA) and BLS, reducing the linear correlation between the data and eliminating redundant fault features. Guo et al. [20] designed a novel recurring BLS fault diagnosis model based on the original BLS. The model inherited the advantages of the BLS and achieved nearly 100% accuracy on two public datasets. The BLS has proven successful for fault diagnosis; however, although the single hidden layer structure of the BLS results in a low computational cost and fast calculation speed, the deep internal features contained in the data have not been sufficiently exploited.

In order to solve these problems, we propose an intelligent framework (LTCN-IBLS) consisting of a 1-D lightweight temporal convolutional network (LTCN) backbone, a two-dimensional (2-D) LTCN backbone, and a BLS with incremental learning (IBLS) classifier to diagnose faults of rotating machinery. The comparison of the LTCN-IBLS and other methods is summarized in Table 1, where the TDFs, TFDFs, and HLSFs denote time-domain features, time–frequency domain features, and high-level semantic features, respectively. The main contributions of this paper can be summarized as follows:(1)The 1-D LTCN and 2-D LTCN are proposed to diagnose faults of rotating machinery. The number of trainable parameters of the LTCN-IBLS is lower than that of many networks that only use 1-D convolution.(2)The temporal features contained in the raw signals and the time–frequency features contained in images transformed by continuous wavelet transform (CWT) are extracted by the 1-D LTCN and 2-D LTCN backbones, respectively. The features are time-constrained, accurately representing the amplitude and frequency variation of the vibration signals of different faults over time. Subsequently, the features of the two domains are fused to obtain distinguishing and comprehensive features for the classifier.(3)The IBLS classifier is adopted to replace the traditional fully connected classification layer, significantly improving the nonlinear mapping ability of the LTCN-IBLS to link the features and fault classes.(4)A detailed comparative analysis is conducted by comparing the proposed framework (LTCN-IBLS) with other state-of-the-art models. Three datasets (two public datasets and one laboratory dataset including compound faults) are employed to verify the effectiveness and lightweight of the framework. Gaussian white noise is introduced in the raw vibration data to evaluate the robustness of the LTCN-IBLS. Ablation experiments are implemented to reveal the contributions of the components of the proposed framework.

The rest of this paper is organized as follows. Section 2 presents the related works. Section 3 describes the details of the proposed LTCN-IBLS framework. Section 4 presents the details of the datasets and experiments and provides the results and discussion. Section 5 concludes the paper.

## 2. Related Works

### 2.1. Continuous Wavelet Transform

Suitable data can significantly improve the performance of DLFD models. With the development of sensor technology, many types of sensors, such as accelerometers and built-in encoders, have been used for mechanical condition monitoring [21]. The temporal signal collected by a sensor during working conditions is complex and non-stationary, complicating the identification of the fault features [11,22]. Therefore, signal-processing algorithms, such as Fourier transform (FT), short-time Fourier transform (STFT), and wavelet transform (WT), are useful for fault diagnosis. FT is a widely used signal-processing algorithm but has several disadvantages [23]. It cannot display the local frequency-domain features of the signal, and the corresponding time-domain information prevents the accurate processing of non-stationary signals. Although STFT can process non-stationary signals, it cannot change the time–frequency resolution during signal processing [24]. The WT has more basic functions and can perform multi-resolution signal analysis; thus, it is a preferred algorithm for fault diagnosis of rotating machinery [25,26]. Therefore, the WT is employed in our work.

CWT is an excellent signal processing tool. Its mathematical expression is as follows:(1)$$cwt\left(\gamma ,\varepsilon \right)=\gamma^{-\frac{1}{2}}\int x\left(t\right)\theta^{*}\left(\frac{t-\varepsilon }{\gamma }\right)dt$$
where *x*(*t*) is the raw 1-D vibration signal. θ and * denote the wavelet mother function and the operator of the complex conjugate, respectively. ε denotes the translation parameter, and γ represents the scale factor. $\gamma^{-1/2}$ is the normalization of energy between each scale so that the transformed signal has the same energy on each scale [27]. The CWT converts the raw time-domain signals into 2-D time–frequency images that can be used as the input of the convolutional layer [28]. The CWT has a fast response speed and provides information with a good balance between the time and frequency resolutions [29,30]. Therefore, we employ the CWT as the signal processing tool to convert the raw data from the time domain into the time–frequency domain.

### 2.2. Temporal Convolutional Network

The TCN, an improved 1-D CNN, was proposed by Bai et al. [31] and has a powerful memory ability to mine historical information from sequential data. The structure of the TCN is shown in Figure 1. It utilizes dilated causal convolution (DCC) and residual learning [32]. The K and D in Figure 1 present the kernel size and dilation, respectively. Unlike the traditional 1-D CNN, the TCN utilizes DCC to capture more useful historical information without stacking too many layers [33]. Furthermore, the DCC results in two other characteristics. First, the current output of the TCN is only affected by the information from the past moment. Second, the output and input of the TCN have the same sequence length. Residual learning enables the TCN to eliminate the impact of gradient explosion or vanishing on the performance of the deep network. TCNs have advantages over recurrent neural networks (RNNs) in concurrent data processing. The performance of TCNs has exceeded RNNs in many fields such as audio synthesis and machine translation [31]. Therefore, TCNs have contributed to researchers removing the dependence of the RNNs in the tasks of sequence modeling and temporal features extraction, which is beneficial to the diversity of solutions.

### 2.3. Broad Learning System with Incremental Learning of Additional Enhancement Nodes

Traditional DL algorithms typically suffer from a complex model structure, causing time-consuming training. Chen and Liu [17] used a random vector functional-link neural network and proposed the BLS to overcome the dependence on the deep structure of the DL algorithms. The incremental learning of additional enhancement nodes (ILAEN) [17], used in this study, adjusts the model to provide better performance without requiring retraining. The structure of the IBLS is shown in Figure 2. The mapped features ($Z_{i}$) and the enhancement nodes ($H_{i}$) are expressed as follows:(2)$$Z_{i}=\varphi \left(XW_{ei}+b_{ei}\right),\;i=1,2,\cdots ,n$$
(3)$$H_{j}=\varepsilon \left(Z^{n}W_{hj}+b_{hj}\right),j=1,2,\cdots ,m$$
(4)$$Output\left(Y\right)=\left[Z^{n}|H_{j}\right]W^{m},j=1,2,\cdots ,m$$
where $Z^{n}\equiv \left[Z_{1},Z_{2},\cdots ,Z_{n}\right]$. X is the input data. $W_{ei}$ and $b_{ei}$ represent the weights and biases of the mapped feature node of the ith set, respectively. $W_{hj}$ and $b_{hj}$ are the weights and biases of the enhancement node of the jth set. $\varphi \left(\text{·}\right)$ and $\varepsilon \left(\cdot \right)$ are activation functions. $W^{m}$ can be obtained from the following equation:(5)$$W^{m}={\left([Z^{n}|H_{j}]\right)}^{ℵ}Y,j=1,2,\cdots ,m$$
where ${\left(\cdot \right)}^{ℵ}$ represents the operation to derive the pseudo-inverse matrix. When the performance of the BLS with the ILAEN is unsatisfactory, the model can be rapidly adjusted by adding additional enhancement nodes. Only the pseudo-inverse matrix of the additional enhancement nodes has to be computed without retraining the entire model. Denote $A^{m}=[Z^{n}|\left[H_{1},H_{2},\cdots ,H_{m}\right]]$ and $F={\left(A^{m}\right)}^{ℵ}H_{m+1}$; the weights ($W^{m+1}$) of additional enhancement nodes are defined in Equations (6) and (7). The derivation process of the IBLS has been described in [17]. The BLS has been widely used in classification and regression tasks due to its advantages of fast training speed and high accuracy. Besides, the BLS enables researchers to model without limiting themselves to deep network structure. However, the accuracy of BLS largely depends on the number of nodes, resulting in redundant nodes and trainable parameters [34].
(6)$$W^{m+1}=\left[\begin{array}{c} W^{m}-FE^{T}Y\\ E^{T}Y \end{array}\right]$$
where
(7)$$E^{T}=\left\{\begin{array}{c} {\left(H_{m+1}-A^{m}F\right)}^{ℵ},\;\;\;\;\;\;\;\;\;\;if\;\left(H_{m+1}-A^{m}F\right)\neq 0\\ {\left(1+F^{T}F\right)}^{-1}B^{T}{\left(A^{m}\right)}^{ℵ},if\;\left(H_{m+1}-A^{m}F\right)=0 \end{array}\right.$$

### 2.4. Evaluation Metrics

Four common evaluation metrics (accuracy, macro-precision (MP), macro-recall (MR), and macro-F_1_ score (MF)) are used to evaluate the model’s classification performance. The accuracy is the ratio of the number of corrected classified samples to the total number of samples [35]. The MP and the MR represent the average number of precision values for each label and the average number of recall values for each label. The MF is the harmonic average of the MP and the MR. The four evaluation metrics are defined as follows:(8)$$accuracy=\frac{TP+TN}{TP+FP+TN+FN}$$
(9)$$MP=\frac{1}{n}\overset{n}{\underset{i=1}{\sum }}P_{i}$$
(10)$$MR=\frac{1}{n}\overset{n}{\underset{i=1}{\sum }}R_{i}$$
(11)$$MF=\frac{2\times MP\times MR}{MP+MR}$$
where $P_{i}$ and $R_{i}$ represent *TP*/(*TP* + *FP*) and *TP*/(*TP* + *FN*), respectively. *TP*, *FP*, *TN*, and *FN* denote the true positive, false positive, true negative, and false negative, respectively.

## 3. Proposed Method

An intelligent fault diagnosis framework (shown in Figure 3a) for rotating machinery (LTCN-IBLS), which has excellent fault diagnosis performance and fewer trainable parameters, is proposed based on the 1-D LTCN, the 2-D LTCN, and the IBLS. In traditional fault diagnosis models, the extracted features must be input into the fully connected classifier for fault identification. However, stacking multiple dense layers is generally required to meet diagnostic requirements and improve accuracy, resulting in excessive trainable parameters for the classifier. Therefore, we replace the dense layers with the IBLS to achieve a lightweight classifier. In addition, we propose a feature extraction stage (stage 1) with a lightweight network structure to perform adaptive representation learning of data before identifying the fault classes to minimize the redundancy of the IBLS nodes and trainable parameters and achieve high accuracy.

In stage 1, two branches are used to obtain comprehensive fault information from time–frequency features and temporal features: a 1-D LTCN backbone (branch 1) and a 2-D LTCN backbone (branch 2). The raw vibration signal is divided into multiple samples without shuffling; each sample contains *N* data points. We consider the following two factors to determine the value of *N*. First, each sample contains enough data points to represent the fault features of a full rotation cycle of the equipment. Second, we minimize *N* to satisfy the first factor and reduce the computational cost. Therefore, the rules for determining *N* can be summarized as follows:(12)$$N=\frac{f_{s}}{s_{r}}$$
where $f_{s}$ is the sampling frequency, and $s_{r}$ is the rotation speed of the equipment.

In branch 1 (shown in Figure 3a), the raw signal is converted into time–frequency images using the CWT. The sampling period of the frequency is the same as that of the vibration signal. The Morlet wavelet is used in our work because its shape is similar to the pulse signal of mechanical faults, facilitating fault diagnosis [4]. A 2-D LTCN backbone with one two-dimensional lightweight DCC (2-D LDCC) block and two LTCN blocks is used to mine distinguishing time–frequency features with time constraints from the CWT images, representing the frequency variation of the vibration signals of different faults. The 2-D LDCC Block (shown in Figure 3b) consists of two group convolution (GConv) layers, two cutting layers, two batch normalization (BN) layers, two ReLU layers, one channel shuffle (CS) layer, and one adaptive max pooling (AMP) layer. The 2-D LTCN Block 1 and 2-D LTCN Block 2 are shown in Figure 3c. They consist of three GConv layers, two cutting layers, two BN layers, two ReLU layers, two CS layers, and two AMP layers. The GConv layers substantially reduce the number of trainable parameters of the 2-D LTCN backbone by grouping input time–frequency features. The CS layers ensure the information flow between different groups by exchanging the features of different groups. The cutting layers remove the frequency features from the feature images in future moments to ensure that the output of the 2-D LTCN backbone has strict time constraints. The AMP 1 of the two 2-D LTCN blocks adjusts the size of the feature images after the cutting layers to obtain a residual connection. The output size of the AMP of the 2-D LDCC block is 25 × 25. The output sizes of the 2-D LTCN block 1 and block 2 are 11 × 11 and 1 × 1, respectively. It is worth noting that the cutting layers are activated only in the direction of the time axis during convolution to ensure that the frequency information at the edge of the feature maps is not lost. The detailed parameters of the 2-D LTCN backbone are summarized in Table 2. In branch 2, a 1-D LTCN backbone with a one-dimensional lightweight DCC (1-D LDCC) block and two 1-D LTCN blocks is utilized to extract the sequential features from the raw data. The 1-D LDCC Block (shown in Figure 3d) is composed of two 1-D GConv layers, two cutting layers, two BN layers, two ReLU layers, one CS layer and one AMP layer. The 1-D LTCN Block 1 and 1-D LTCN Block 2 are shown in Figure 3e. They consist of three 1-D GConv layers, two cutting layers, two BN layers, two ReLU layers, two CS layers, and one AMP layer. Similar to the function in the 2-D LTCN backbone, the 1-D GConv layers and CS layers enable the 1-D LTCN backbone to reduce the number of trainable parameters by grouping 1-D temporal features and achieving information communication between different groups. The cutting layers remove information on future moments contained in 1-D temporal features so that the current output of the 1-D LTCN backbone is only affected by information from past moments. The AMP layers of the 1-D LDCC Block and 1-D LTCN Block 1 reduce the dimension of the blocks’ output by half. The output size of the AMP layer in the 1-D LTCN Block 2 is 1. The detailed parameters of the 1-D LTCN backbone are presented in Table 2. The 2-D and 1-D LTCN backbones ensure that the feature extraction stage of LTCN-IBLS has sufficient receptive fields to extract representative temporal features and high-level semantic information from the time–frequency images and raw vibration signals while minimizing the number of trainable parameters.

The cross-entropy function (Equation (13)) is used as the loss function for training the 2-D LTCN backbone and the 2-D LTCN backbone. The time–frequency features (F_1_) extracted by the 2-D LTCN backbone and the sequential features (F_2_) extracted by the 1-D LTCN backbone are fused into fused feature vectors (Fu).
(13)$$loss=-\overset{n}{\underset{i=1}{\sum }}l_{i}\log\left(y_{i}\right)$$

After completing the representation learning of data in stage 1, we input the extracted fault features into the fault diagnosis stage (stage 2) and adopt the IBLS to replace the fully connected classifier to improve the ability to establish complex nonlinear relationships between features and fault classes. In stage 2, the IBLS receives the fused feature vectors from the feature extraction stage and diagnoses the faults. Z, H, and A in Figure 3a denote the feature mapping nodes, enhancement nodes, and additional enhancement nodes, respectively. The feature mapping nodes and the enhancement nodes are defined as follows:(14)$$Z_{i}=\varphi \left(FuW_{ei}+b_{ei}\right),i=1,2,\cdots ,r$$
(15)$$H_{j}=\varepsilon \left(Z^{r}W_{hj}+b_{hj}\right),j=1,2,\cdots ,s$$
where $Z^{r}\equiv \left[Z_{1},Z_{2},\cdots ,Z_{r}\right]$. $\varphi \left(\cdot \right)$ and $\varepsilon \left(\cdot \right)$ are Tanh functions. $W_{ei}$ and $b_{ei}$ denote the weights and the biases of the ith set of the feature mapping nodes, respectively. They are randomly generated in the range of −1 to 1. $W_{hi}$ and $b_{hi}$ represent the weights and the biases of the jth set of the enhancement node and are randomly generated in the range of −1 to 1. r and s are 1. Equation (5) is used to define the input weights of the output layer in Equation (16). When the faults diagnosis results are unsatisfactory, additional enhancement nodes can be added to improve the results. According to Equations (6) and (7), denoting $F={\left(A^{s}\right)}^{ℵ}H_{s+1}$, the weights ($W^{s+1}$) between the additional enhancement nodes and the output layer are obtained using Equations (17) and (18).
(16)$$W^{s}={\left(Z_{r}|H_{j}\right)}^{ℵ}Y,j=1,2,\cdots ,s$$
(17)$$W^{s+1}=\left[\begin{array}{c} W^{s}-FE^{T}Y\\ E^{T}Y \end{array}\right]$$
where
(18)$$E^{T}=\left\{\begin{array}{c} {\left(H_{s+1}-A^{s}F\right)}^{ℵ},\;\;\;\;\;\;\;\;\;\;\;if\;\left(H_{s+1}-A^{s}F\right)\neq 0\\ {\left(1+F^{T}F\right)}^{-1}B^{T}{\left(A^{s}\right)}^{ℵ},if\;\left(H_{m+1}-A^{s}F\right)=0 \end{array}\right.$$

${\left(\cdot \right)}^{ℵ}$ represents the operation to acquire the pseudo-inverse matrix. Y denotes the fault labels.

The proposed LTCN-IBLS framework is programmed using PyTorch 1.7.1. The framework’s workflow is summarized in Figure 4.

## 4. Results and Discussion

We used three datasets (two public datasets and one laboratory dataset) to verify the fault diagnosis performance of the proposed LTCN-IBLS framework and compared it with six other powerful models (R-O-IBLS, Deep Convolutional Neural Networks with Wide First-layer Kernels (WDCNN) [36], 1-D deep CNN (1-D DCNN) [9], Deep ResNet [10], CWT-CNN [29], and AlexNet [37]). In the R-O-IBLS, the 2-D LTCN backbone shown in Figure 3a was replaced by a 1-D ResNet with the same network depth. Furthermore, ablation experiments were conducted to evaluate the performance of the proposed framework. The OLTCN-IBLS does not have the 2-D LTCN backbone shown in Figure 3a, and the TLTCN-IBLS does not contain the 1-D LTCN backbone shown in Figure 3a. In the OLTCN-TLTCN, the IBLS classification module shown in Figure 3a was replaced by a fully connected layer. After every five epochs during the training process, the learning rates were multiplied by 0.9. The batch sizes were 0.1 times the number of training samples. When the loss changes of the validation set were less than 0.01, it was assumed the model had converged.

Gaussian white noise with a signal-to-noise ratio (SNR) equal to 0 was introduced into the datasets to verify the models’ robustness. Five-fold cross-validation was utilized to assess the performance of the training process. Four evaluation metrics (accuracy, MR, MP, and MF) and the number of trainable parameters were employed to assess the results of the experiments. The objective was to obtain high evaluation metrics while minimizing the number of trainable parameters. All the experimental results are the average value of ten repeated trials to reduce the effect of randomness. All the experiments were implemented on a computer with two NVIDIA Tesla V100s graphics cards, and programming was performed with PyTorch 1.7.1.

### 4.1. Case 1: Case Western Reserve University (CWRU) Dataset

#### 4.1.1. Dataset Description

The CWRU dataset [38] provided by the bearing center at CWRU is a well-known and representative bearing dataset that contains sufficient standard experimental data. It has been used extensively for fault diagnosis research [39]. The sampling frequency in the experiments is 12 kHz. The equipment used in the experiment is shown in Figure 5. Three types of faults (inner race defect, ball defect, and outer race defect) were created in the experimental bearing components using electro-discharge machining. Each fault class has bearing components with three diameters (7 mils, 14 mils, and 21 mils). The experiments were conducted under four different loads (0 hp, 1 hp, 2 hp, and 3 hp). Therefore, the dataset had ten fault classes (one healthy class and 9 fault classes). The minimum rotation speed of the facility under the four different loads was 1730 rpm. According to (12), the number of data points per sample was 417. Without shuffling or using duplicate data, 70% of the data was used for training, and 30% was used for testing. The details of the dataset are summarized in Table 3.

#### 4.1.2. Diagnosis Results of the Experiments for Case 1

The mean values (MV) and standard deviations (SD) of the ablation experiments’ results are summarized in Table 4 representing the contributions of the components of the proposed LTCN-IBLS framework. The LTCN-IBLS framework achieved the best performances based on all four evaluation metrics, with an accuracy of 0.9875, an MP of 0.9882, an MR of 0.9873, and an MF of 0.9872. The OLTCN-IBLS obtained the worst performance with an accuracy of 0.9483, an MP of 0.9495, an MR of 0.9488, and an MF of 0.9482, worse than the performance of the TLTCN-IBLS. The diagnosis results of the OLTCN-IBLS had high dispersions, and the SDs of all metrics exceeded 0.0050. The accuracy, MP, MR, and MF of the LTCN-IBLS were 0.0095, 0.0094, 0.0091, and 0.0094 higher than those of the OLTCN-TLTCN. Table 5 displays the average results of the four evaluation metrics and the number of trainable parameters for the different models for case 1. The proposed framework had the highest values of the evaluation metrics and the lowest parameter number (0.0149 mega (M)). The values of the four evaluation metrics of the LTCN-IBLS were 0.0560, 0.0426, 0.0560, and 0.0584 higher than those of the Deep ResNet, and the number of trainable parameters of the LTCN-BLS was 57.0299 M lower than that of the AlexNet.

#### 4.1.3. Diagnosis Results of the Experiments for Case 1 under Noisy Conditions

Table 6 presents the average results of the ablation experiments under noisy conditions. Only the LTCN-IBLS’s four evaluation metrics exceeded 0.9300, with an accuracy of 0.9409, an MP of 0.9458, an MR of 0.9402, and an MF of 0.9377. In addition, the SDs of the LTCN-IBLS’s metrics were the lowest, indicating low dispersion of the LTCN-IBLS’s diagnosis results. The TLTCN-IBLS had the worst diagnosis performance; the four metrics were lower than 0.8600. The OLTCN-TLTCN achieved the second-best results with an accuracy of 0.9003, an MP of 0.9220, an MR of 0.9005, and an MF of 0.8883. These values were 0.0324, 0.0518, 0.0323, and 0.0227 higher than those of the OLTCN-IBLS. The average results of the comparative experiments under noisy conditions for different models are shown in Table 7. The LTCN-IBLS achieved the best diagnosis results and the lowest number of trainable parameters. The values of the four evaluation metrics of the LTCN-IBLS were 0.1446, 0.1365, 0.1444, and 0.1492 higher than those of the CWT-CNN. The dispersion of the CWT-CNN’s MP was high, with an SD of 0.0104.

### 4.2. Case 2: Intelligent Maintenance Systems Bearing Dataset

#### 4.2.1. Dataset Description

The Intelligent Maintenance Systems (IMS) bearing dataset [40,41] is a public dataset created by the Prognostics Center of Excellence at the University of Cincinnati. A bearing test rig (shown in Figure 6) was used to perform a bearing run-to-failure test to collect the bearing data. The four test bearings were installed on a shaft driven by an AC motor with a rotation speed of 2000 rpm. A constant load of 6000 lbs was added to the shaft, and the sampling rate was 20 kHz [41]. After a certain amount of metal debris had adhered to the magnetic plug, an advanced stage of degradation was reached, and the test was stopped. The samples of the three fault classes (healthy, inner race defect, and roller element defect) were obtained from the data files collected on 25 November 2003 [40], and those of the outer race defect were obtained from the data files collected on 19 February 2004 [40]. According to [12], the number of data points of each sample was 600. Without shuffling the data, we used 357,000 data points from each fault class as the training dataset and 153,000 as the test dataset. The details of the dataset are presented in Table 8.

#### 4.2.2. Diagnosis Results of the Comparative Experiments for Case 2

As shown in Table 9, the LTCN-IBLS achieved the best performance in the ablation experiments based on all four evaluation metrics, with an accuracy of 0.9999, an MR of 0.9999, an MP of 0.9999, and an MF of 0.9999. The OLTCN-TLTCN achieved the second-best performance, and all metric values were 0.9961. The OLTCN-IBLS and TLTCN-IBLS had similar values of the four metrics, ranging from 0.9860 to 0.9880. The overall fault diagnosis results of the different models for case 2 are summarized in Table 10. The LTCN-IBLS achieved the highest performance, with the highest values of the four performance metrics and the lowest number of trainable parameters. The values of the LTCN-IBLS were 0.0141, 0.0131, 0.0139, and 0.0141 higher than those of the AlexNet; the parameter number of the LTCN-BLS was 57.0035 M lower than that of the AlexNet.

#### 4.2.3. Diagnosis Results of the Experiments for Case 2 under Noisy Conditions

Table 11 shows the average diagnosis results of the ablation experiments under noisy conditions. The LTCN-IBLS achieved the best performance with an accuracy of 0.9753, an MP of 0.9752, an MR of 0.9753, and an MF of 0.9752. These values were 0.0361, 0.0253, 0.0373, and 0.0427 higher than those of the OLTCN-TLTCN. The OLTCN-IBLS obtained unsatisfactory diagnosis results with all four metrics lower than 0.9227. The average diagnosis results of the comparative experiments under noisy conditions are presented in Table 12. The LTCN-IBLS achieved the best performance. The MF of the 1-D DCNN had a low dispersion with an SD of 0.0005. However, the four metrics of the 1-D DCNN were 0.0062, 0.0031, 0.0056, and 0.0044 lower than those of the LTCN-IBLS. The parameter number of the 1-D DCNN was 0.0496 M higher than that of the LTCN-IBLS.

### 4.3. Case 3: Gear and Bearing Dataset

#### 4.3.1. Dataset Description

This dataset consisted of vibration data obtained from an experimental platform of a multi-stage centrifugal air compressor unit (MCACU). It contained several common faults of gears and bearings. Five fault components (three bearings with inner race defects, outer race defects, missing balls, and two gears with missing teeth) were used in our experiments, and single faults and compound faults were considered. The details of the five fault components are displayed in Figure 7 and Table 13. The components and parameters of the experimental platform are shown in Figure 8a and Table 14, respectively. Figure 8b shows the details of the parts inside the red rectangle in Figure 8a, including the positions of the fault components and the velocity sensor used for data sampling.

A data acquisition system (Figure 8c) was used to acquire and store the vibration data. The system includes a velocity sensor (Figure 8b), a signal conditioning module (not shown), a server, a data collector module, and a monitor. Two independent experiments with the MCACU were conducted in the same environment (temperature: 20 ± 1 °C, relative humidity: 65 ± 5%) to obtain training samples and test samples. The gross errors were removed during the data collection. The sampling rate was 1024 Hz, and the data were saved every 20 s (8192 data points were saved each time). The data of the first 10 s of each experiment were not saved so that only stable operation data were used in the analysis. According to [12], each sample contained 248 (62 × 4) data points. For each fault class, 347,200 data points were collected, and 1400 training samples were obtained; 148,800 data points were collected, and 600 test samples were obtained. The details of the dataset are summarized in Table 15.

#### 4.3.2. Diagnosis Results of the Comparative Experiments for Case 3

The average results of the ablation experiments for case 3 are summarized in Table 16. The LTCN-IBLS achieved the best fault diagnosis performance, with an accuracy of 0.9800, an MP of 0.9806, an MR of 0.9800, and an MF of 0.9797. The second-best performance was achieved by the OLTCN-TLTCN, with accuracy, MP, MR, and MF values of 0.9726, 0.9732, 0.9726, and 0.9725, respectively. The OLTCN-IBLS and TLTCN-IBLS had similar diagnosis performance, and the four metrics ranged from 0.9390 to 0.9470. The average diagnosis results of the different models and the number of trainable parameters for case 3 are listed in Table 17. The LTCN-IBLS achieved the best diagnosis performance and the lowest number of trainable parameters (0.0101 M). The values of the LTCN-IBLS were 0.0404, 0.0339, 0.0404, and 0.0411 higher than those of the CWT-CNN, and the number of trainable parameters of the LTCN-BLS was 57.0183 M lower than those of the AlexNet. The confusion matrix of LTCN-IBLS for the test dataset is shown in Figure 9. The samples of the healthy condition (label 0) and single fault (label 1) are all correctly classified. Some samples of compound faults are incorrectly classified, and 60 samples with label 2 are misclassified.

#### 4.3.3. Diagnosis Results of the Experiments for Case 3 under Noisy Conditions

The ablation experiments’ results under noisy conditions are shown in Table 18. The LTCN-IBLS achieved the best diagnostic performance with an accuracy of 0.9158, an MP of 0.9235, an MR of 0.9158, and an MF of 0.9148. The OLTCN-IBLS and TLTCN-IBLS had similar experimental results, with the four metrics ranging from 0.7845 to 0.8098. The values of the OLTCN-TLTCN’s results were 0.0248, 0.0112, 0.0238, and 0.0322 lower than those of the LTCN-IBLS. Table 19 displays the average results of the comparative experiments. The LTCN-IBLS achieved the best evaluation metrics (accuracy, MP, MR, and MF), which were 0.0553, 0.0548, 0.0553, and 0.0580 higher than those of the CWT-CNN. The 1-D DCNN’s metrics obtained the second-highest accuracy, MR and MF. However, the values of the four metrics (accuracy, MP, MR, and MF) of 1-D DCNN were 0.0111, 0.0166, 0.0113, and 0.0112 lower than those of the LTCN-IBLS. The confusion matrix of the LTCN-IBLS framework is presented in Figure 10. All samples of the healthy condition (label 0) are all correctly classified, and only one sample of the single fault is misclassified. However, there are many misclassification cases in the samples of the compound faults (especially label 2 and label 4).

### 4.4. Discussion

The proposed lightweight fault diagnosis framework for rotating machinery exhibited an outstanding performance for fault classification. The effectiveness of the proposed framework was verified by two experiments: (1) ablation experiments were implemented to reveal the contributions of the framework’s components. (2) Comparative experiments were conducted to compare the performance of the proposed framework with other models. The robustness of models was evaluated comprehensively using three datasets under noisy conditions.

The results of the ablation experiments for the three cases indicate that the fault diagnosis performance of the LTCN-IBLS was significantly better than that of the OLTCN-IBLS and TLTCN-IBLS, demonstrating that the time–frequency domain information and sequential features provided non-negligible contributions to the fault diagnosis. We found that the fused features obtained from the feature extraction stage of the proposed framework provided a comprehensive and advanced feature representation of the rotating machinery faults. This approach substantially improved the fault diagnosis performance of the proposed framework. In addition, the results (especially Table 9 and Table 18) of the ablation experiments showed that the temporal features and time–frequency features were almost equally important for fault diagnosis. The LTCN-IBLS outperformed the OLTCN-TLTCN in all three cases, indicating that the IBLS has a stronger classification ability than the traditional classifier with a fully connected layer. Meanwhile, ablation experiments were conducted using the three datasets with Gaussian white noise (SNR = 0). The LTCN-IBLS achieved the best performance, further demonstrating the rationality of the framework.

In the comparative experiments, the proposed LTCN-IBLS framework was compared with other state-of-the-art models on three datasets. The results showed that the LTCN-IBLS framework had an outstanding performance for fault diagnosis and the number of trainable parameters in the three cases. The proposed 2-D LTCN backbone in the proposed framework provided sufficient receptive fields to extract the global and high-level semantic information from the time–frequency images of the rotating machinery faults, enabling the framework to capture the changing trend of the frequency intensity over time. The proposed 1-D LTCN backbone had a strong temporal feature extraction ability to capture the amplitude variation over time hidden in the raw vibration data. The LTCN-IBLS showed better diagnostic performance than the R-O-IBLS in the three cases, indicating that the time–frequency features extracted by the 2-D LTCN backbone contained information indispensable for accurate fault identification. The features extracted by the 2-D LTCN and 1-D LTCN had higher time constraints than those extracted by the other models (WDCNN, 1-D DCNN, Deep ResNet, CWT-CNN, and AlexNet). The proposed strategy ensured that the feature representations corresponding to different fault classes could be distinguished, enabling the classifier to accurately identify the fault categories. Besides, unlike the traditional dense channels of the TCN, the lightweight 2-D LTCN and 1-D LTCN backbones substantially reduced the number of trainable parameters, lowering the complexity of the network structure. The experimental results of the three cases demonstrated that the introduction of noise degraded the fault diagnosis performance of all models. The proposed LTCN-IBLS still achieved the best performance under noisy conditions, indicating that it has higher robustness than other comparable models. As shown in Table 20, a comparison of the convergence time of the training process (CTTP) was performed for models (CWT-CNN, AlexNet, and LTCN-IBLS) with 2-D data processing capability. The proposed LTCN-IBLS achieved the fastest convergence speed in the training process of the three datasets. It is worth mentioning that the input size of the CWT-CNN is smaller than that of LTCN-IBLS, indicating that the LTCN-IBLS has a better training performance.

## 5. Conclusions

In this study, we proposed an intelligent framework (called LTCN-IBLS) consisting of a feature extraction stage and a fault identification stage for data representation learning and faults diagnosis. In the feature extraction stage, the 1-D LTCN and 2-D LTCN backbone were used to extract the time-dependent information, including the time–frequency features and temporal features. Information from the future input in the direction of time axis was removed, while the complete frequency edge and corner information was retained during the extraction of the time–frequency features. The time–frequency features were fused with the temporal features to obtain more in-depth and high-quality information on the fault features to improve the fault identification performance of the IBLS classifier in the fault identification stage. The IBLS classifier established an accurate mapping relationship between the fused features and the fault categories. It exhibited better nonlinear mapping ability than the fully connected layers. Ablation experiments demonstrated the rationality and contributions of the framework’s components. Under non-noisy conditions, the MVs of the accuracy, MP, MR, and MF of the LTCN-IBLS were up to 0.0560, 0.0426, 0.0560, and 0.0584 higher than those of the comparable models. Under noisy conditions, the MVs of the accuracy, MP, MR, and MF of the LTCN-IBLS were up to 0.1446, 0.1365, 0.1444, and 0.1492 higher than those of the comparable models. The LTCN-IBLS had the lowest number of trainable parameters (≤0.0165 M) among all models. The experimental results prove that the proposed framework possesses effectiveness, lightweight, and robustness for fault diagnosis.

This study provided insights and solutions for establishing lightweight neural network models to diagnose faults of rotating machinery, minimizing manual intervention. However, due to the large amount of computation in processing 2-D data, our proposed model is slower than many other models with 1-D input, although it has a low parameter number. Figure 9 and Figure 10 indicate that it is more difficult to diagnose compound faults than single faults. Therefore, we will focus on more effective intelligent algorithms for improving the calculation speed of the models and diagnosing the compound faults of rotating machinery in future studies.

## Figures and Tables

**Figure 1 sensors-23-05642-f001:**
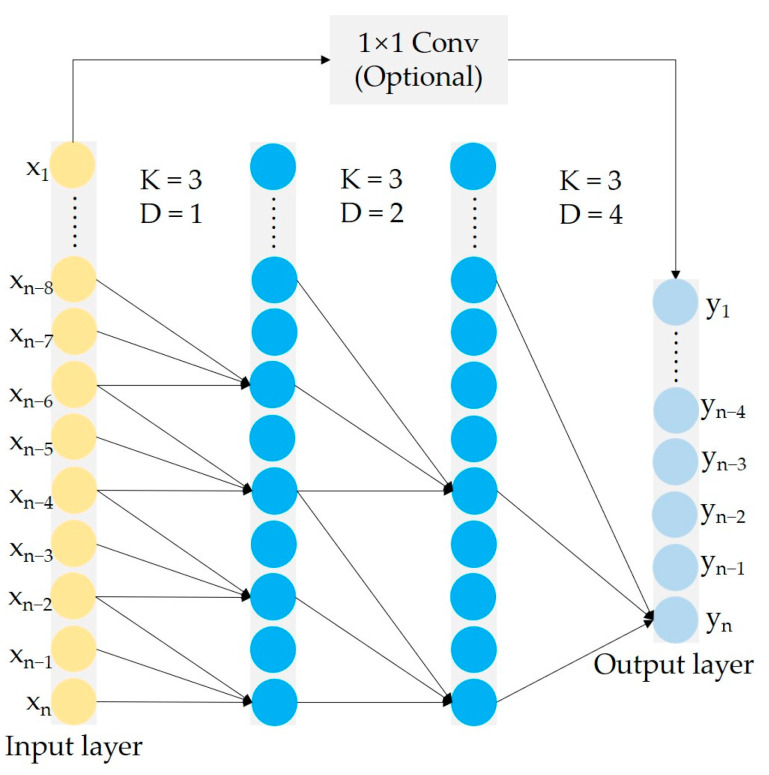
The structure diagram of the TCN.

**Figure 2 sensors-23-05642-f002:**
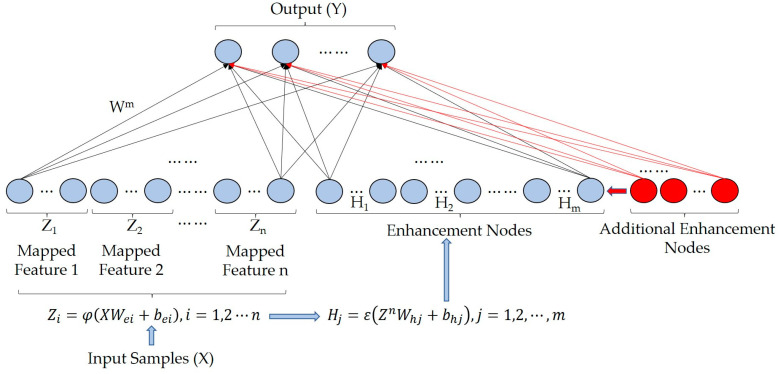
The structure diagram of the BLS with the ILAEN.

**Figure 3 sensors-23-05642-f003:**
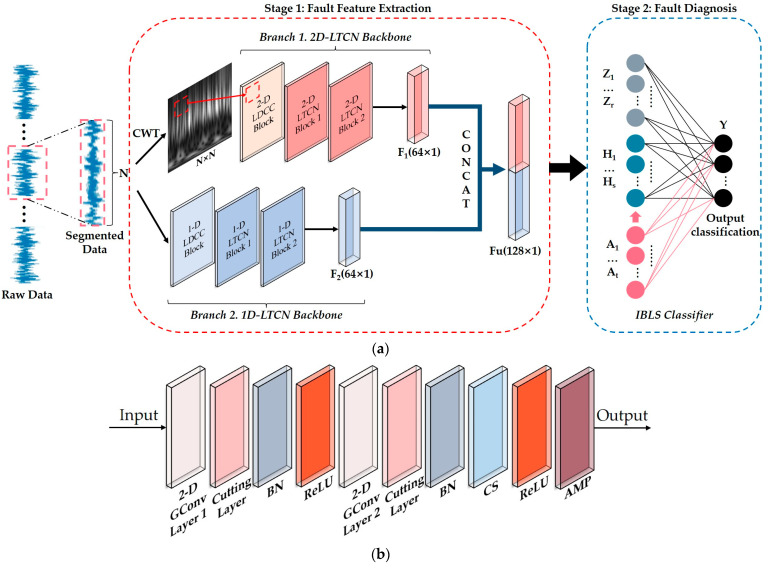
The process diagram of the proposed framework for fault diagnosis. (**a**) is the fault diagnosis process. (**b**) is the structure diagram of the 2-D LDCC Block shown in (**a**). (**c**) is the structure diagram of the 2-D LTCN Block 1 and 2-D LTCN Block 2 shown in (**a**). (**d**) is the structure diagram of the 1-D LDCC Block shown in (**a**). (**e**) is the structure diagram of the 1-D LTCN Block 1 and 1-D LTCN Block 2 shown in (**a**).

**Figure 4 sensors-23-05642-f004:**
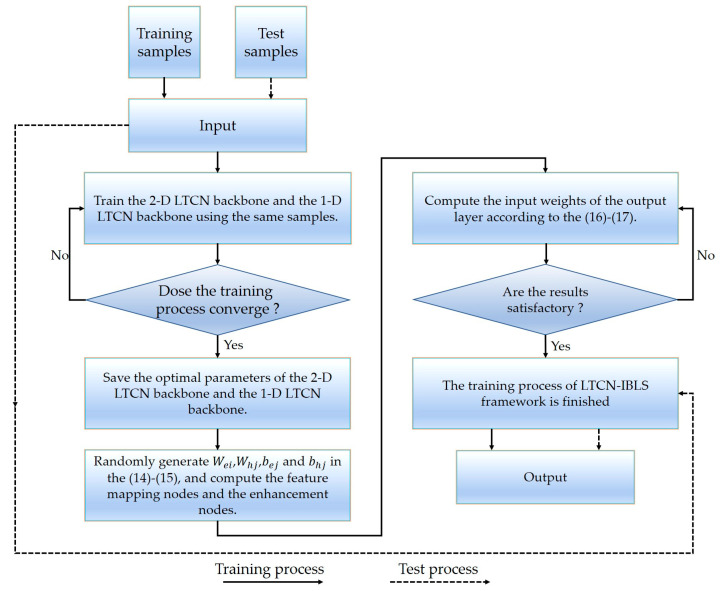
The workflow of the LTCN-IBLS framework.

**Figure 5 sensors-23-05642-f005:**
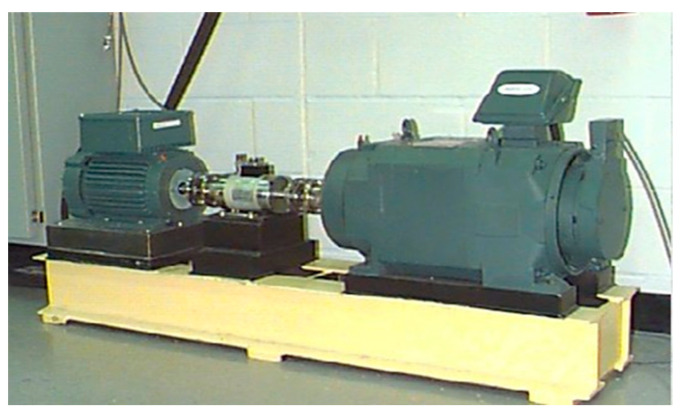
The experimental equipment used to create the CWRU dataset.

**Figure 6 sensors-23-05642-f006:**
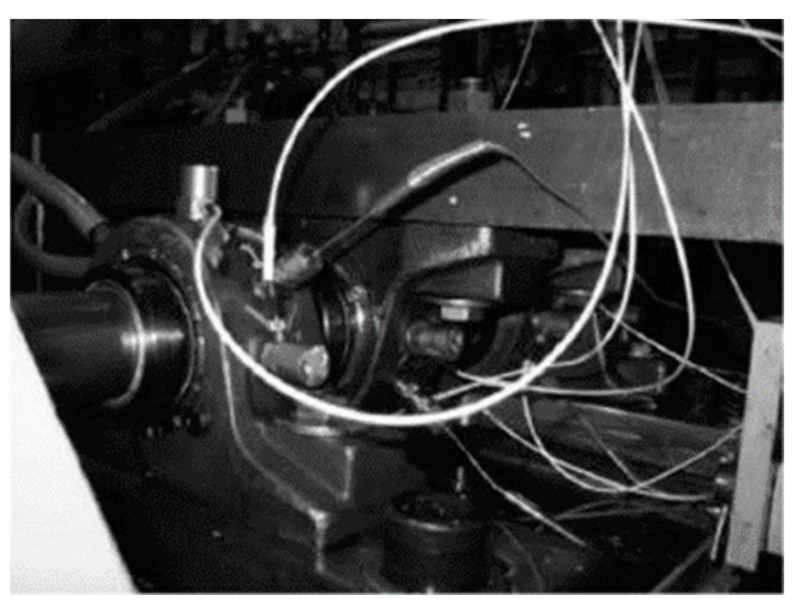
The experimental bearing test rig used to create the IMS dataset.

**Figure 7 sensors-23-05642-f007:**
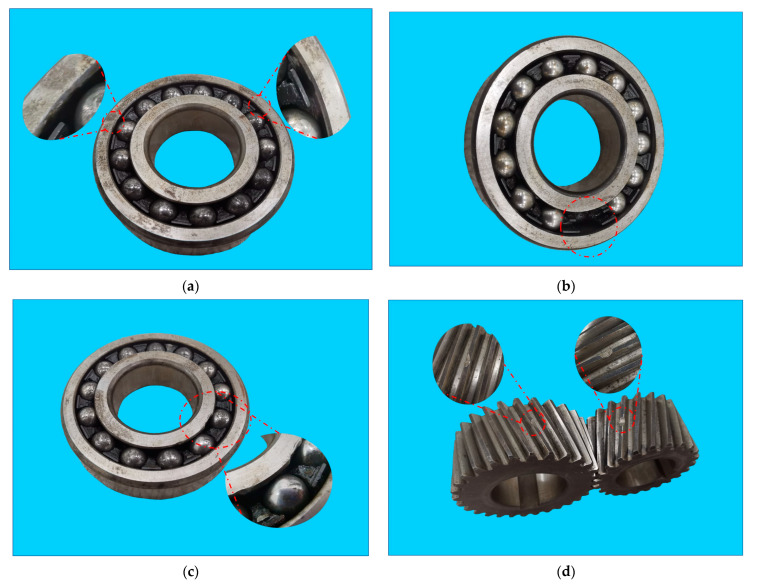
Photos of the five rolling components. (**a**–**c**) show three bearings used in case 3 with an outer race defect, missing ball, and inner race defect, respectively. (**d**) shows two gears with missing teeth used in case 3.

**Figure 8 sensors-23-05642-f008:**
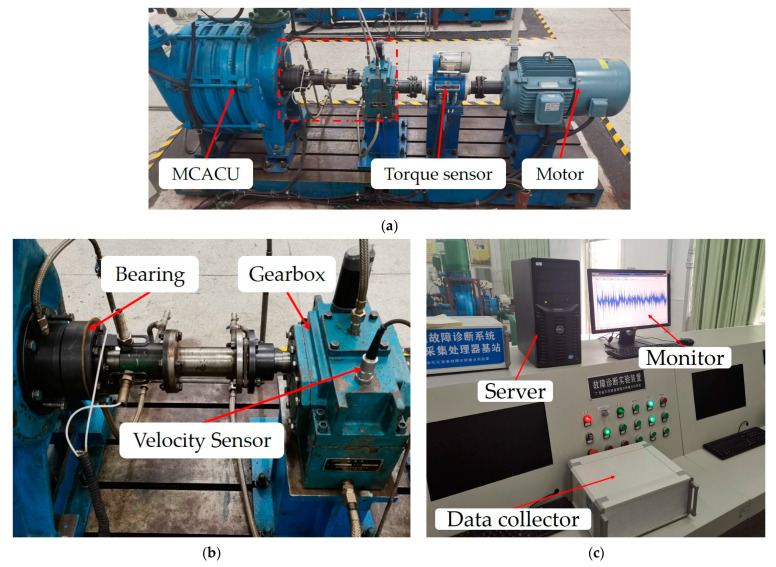
The photos of the experimental test platform used to create the laboratory dataset using the MCACU rig. (**a**,**b**) show the MCACU experimental platform. (**c**) is the data acquisition system.

**Figure 9 sensors-23-05642-f009:**
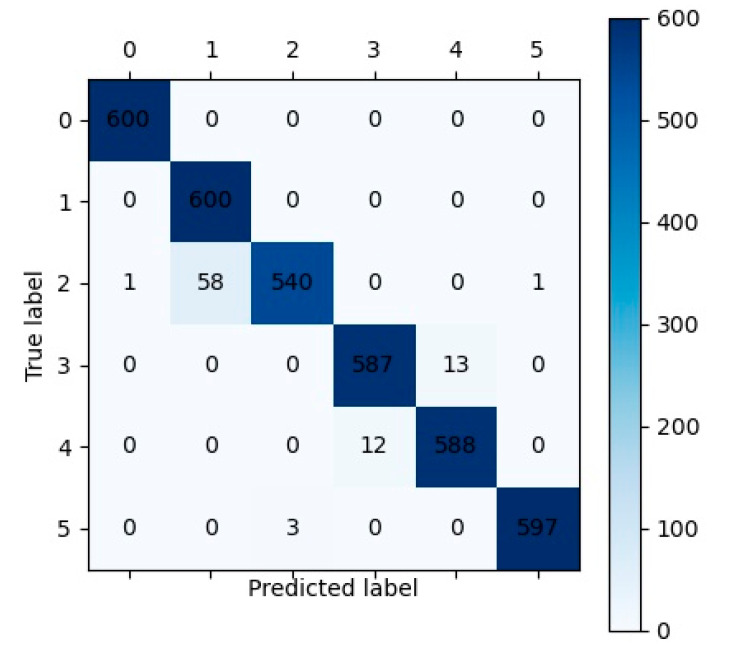
The confusion matrix of the LTCN-IBLS framework for the test dataset of case 3.

**Figure 10 sensors-23-05642-f010:**
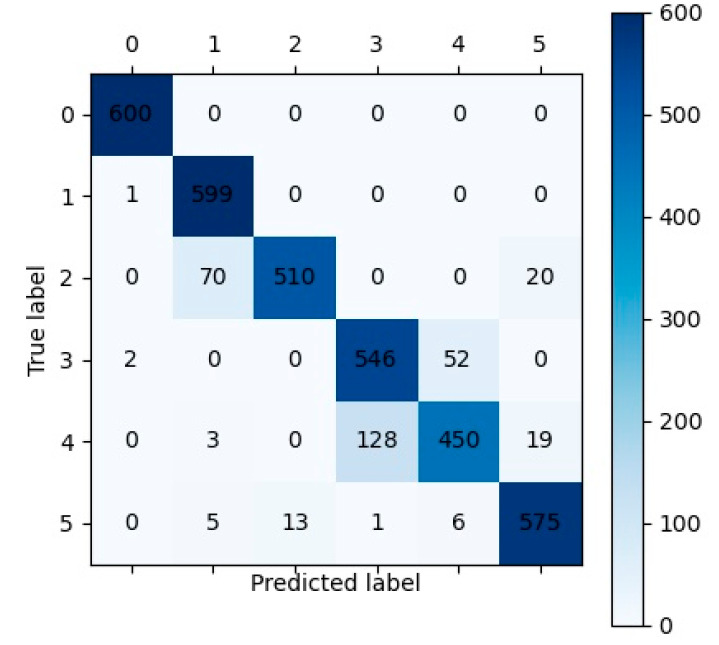
The confusion matrix of the LTCN-IBLS framework for the test dataset of case 3 under noisy conditions.

**Table 1 sensors-23-05642-t001:** Comparison of the LTCN-IBLS and other methods.

Methods	Extract TDFs	Extract TFDFs	Extract HLSFs	Summary
References [9,10]	Yes	No	Yes	Susceptible to signal non-stationarity
References [13,14]	No	Yes	Yes	Susceptible to information loss caused by domain transformation
References [18,19,20]	Yes	No	No	Difficult to extract HLSFs
Our proposed method	Yes	Yes	Yes	Extract HLSFs and comprehensive features containing time-domain and time–frequency domain information

**Table 2 sensors-23-05642-t002:** The parameters of the 2-D LTCN backbone and the 1-D LTCN backbone.

Name		Input Channel	Output Channel	Kernel Size	Stride	Pad	Group	Dilation
2-D LDCC Block	GConv Layer 1	3	8	5	2	4	1	1
	GConv Layer 2	8	8	5	2	4	8	1
2-D LTCN Block 1	GConv Layer 1	8	32	3	1	4	8	2
	GConv Layer 2	32	32	3	1	4	32	2
	GConv Layer 3	8	32	1	1	0	8	1
2-D LTCN Block 2	GConv Layer 1	32	64	3	1	8	32	4
	GConv Layer 2	64	64	3	1	8	64	4
	GConv Layer 3	32	64	1	1	0	32	1
1-D LDCC Block	GConv Layer 1	1	8	16	2	15	1	1
	GConv Layer 2	8	8	16	2	15	8	1
1-D LTCN Block 1	GConv Layer 1	8	32	2	1	2	8	2
	GConv Layer 2	32	32	2	1	2	32	2
	GConv Layer 3	8	32	1	1	0	8	1
1-D LTCN Block 2	GConv Layer 1	32	64	2	1	4	32	4
	GConv Layer 2	64	64	2	1	4	64	4
	GConv Layer 3	32	64	1	1	0	32	1

**Table 3 sensors-23-05642-t003:** Dataset details of case 1.

Label	Load (hp)	Fault Class	Diameter (mil)	Training Samples/Test Samples
0	0/1/2/3	Health	-	427/183
1		Inner Race Defect	7	427/183
2		Ball Defect	14	427/183
3		Outer Race Defect	21	427/183
4		Inner Race Defect	7	427/183
5		Ball Defect	14	427/183
6		Outer Race Defect	21	427/183
7		Inner Race Defect	7	427/183
8		Ball Defect	14	427/183
9		Outer Race Defect	21	427/183

**Table 4 sensors-23-05642-t004:** Average results of the ablation experiments for case 1.

Model	Accuracy		MP		MR		MF	
	MV	SD	MV	SD	MV	SD	MV	SD
OLTCN-IBLS	0.9483	0.0053	0.9495	0.0055	0.9488	0.0053	0.9482	0.0054
TLTCN-IBLS	0.9583	0.0036	0.9606	0.0031	0.9590	0.0038	0.9580	0.0038
OLTCN-TLTCN	0.9780	0.0042	0.9788	0.0039	0.9782	0.0040	0.9778	0.0041
LTCN-IBLS	0.9875	0.0034	0.9882	0.0030	0.9873	0.0033	0.9872	0.0034

**Table 5 sensors-23-05642-t005:** Average diagnosis results of different models for case 1.

Model	Accuracy		MP		MR		MF		Number of Trainable Parameters (M)
	MV	SD	MV	SD	MV	SD	MV	SD	
R-O-IBLS	0.9419	0.0067	0.9568	0.0021	0.9419	0.0067	0.9424	0.0061	0.0349
WDCNN [36]	0.9602	0.0020	0.9616	0.0018	0.9608	0.0024	0.9595	0.0023	0.0155
1-D DCNN [9]	0.9741	0.0052	0.9750	0.0052	0.9746	0.0050	0.9736	0.0052	0.0433
Deep ResNet [10]	0.9315	0.0008	0.9456	0.0009	0.9313	0.0009	0.9288	0.0011	0.0208
CWT-CNN [29]	0.9406	0.0026	0.9512	0.0017	0.9400	0.0030	0.9348	0.0026	5.4073
AlexNet [37]	0.9582	0.0030	0.9618	0.0020	0.9582	0.0029	0.9559	0.0036	57.0448
LTCN-IBLS	0.9875	0.0034	0.9882	0.0030	0.9873	0.0033	0.9872	0.0034	0.0149

**Table 6 sensors-23-05642-t006:** Average results of the ablation experiments for case 1 under noisy conditions.

Model	Accuracy		MP		MR		MF	
	MV	SD	MV	SD	MV	SD	MV	SD
OLTCN-IBLS	0.8679	0.0034	0.8702	0.0029	0.8682	0.0033	0.8656	0.0035
TLTCN-IBLS	0.8515	0.0040	0.8560	0.0041	0.8522	0.0040	0.8453	0.0060
OLTCN-TLTCN	0.9003	0.0044	0.9220	0.0042	0.9005	0.0043	0.8883	0.0076
LTCN-IBLS	0.9409	0.0017	0.9458	0.0021	0.9402	0.0016	0.9377	0.0022

**Table 7 sensors-23-05642-t007:** Average diagnosis results of different models for case 1 under noisy conditions.

Model	Accuracy		MP		MR		MF		Number of Trainable Parameters (M)
	MV	SD	MV	SD	MV	SD	MV	SD	
R-O-IBLS	0.9110	0.0020	0.9131	0.0030	0.9113	0.0023	0.9100	0.0018	0.0349
WDCNN [36]	0.8913	0.0040	0.8979	0.0026	0.8908	0.0047	0.8879	0.0040	0.0155
1-D DCNN [9]	0.8566	0.0035	0.8684	0.0045	0.8561	0.0034	0.8529	0.0031	0.0433
Deep ResNet [10]	0.8996	0.0036	0.9049	0.0023	0.9010	0.0040	0.8977	0.0044	0.0208
CWT-CNN [29]	0.7963	0.0047	0.8093	0.0104	0.7958	0.0048	0.7885	0.0065	5.4073
AlexNet [37]	0.8569	0.0045	0.8683	0.0031	0.8561	0.0046	0.8505	0.0047	57.0448
LTCN-IBLS	0.9409	0.0017	0.9458	0.0021	0.9402	0.0016	0.9377	0.0022	0.0149

**Table 8 sensors-23-05642-t008:** Dataset details of case 2.

Label	Fault Class	Training Samples/Test Samples
0	Health	595/255
1	Inner Race Defect	595/255
2	Roller Element Defect	595/255
3	Outer Race Defect	595/255

**Table 9 sensors-23-05642-t009:** Average results of the ablation experiments for case 2.

Model	Accuracy		MP		MR		MF	
	MV	SD	MV	SD	MV	SD	MV	SD
OLTCN-IBLS	0.9869	0.0045	0.9880	0.0041	0.9869	0.0044	0.9869	0.0046
TLTCN-IBLS	0.9860	0.0029	0.9863	0.0027	0.9860	0.0029	0.9860	0.0029
OLTCN-TLTCN	0.9961	0.0012	0.9961	0.0012	0.9961	0.0012	0.9961	0.0012
LTCN-IBLS	0.9999	0.0002	0.9999	0.0002	0.9999	0.0002	0.9999	0.0002

**Table 10 sensors-23-05642-t010:** Average diagnosis results of different models for case 2.

Model	Accuracy		MP		MR		MF		Number of Trainable Parameters (M)
	MV	SD	MV	SD	MV	SD	MV	SD	
R-O-IBLS	0.9975	0.0042	0.9976	0.0040	0.9975	0.0043	0.9975	0.0042	0.0294
WDCNN [36]	0.9985	0.0026	0.9986	0.0025	0.9986	0.0025	0.9986	0.0025	0.0174
1-D DCNN [9]	0.9998	0.0004	0.9998	0.0003	0.9998	0.0004	0.9998	0.0004	0.0661
Deep ResNet [10]	0.9988	0.0021	0.9990	0.0018	0.9989	0.0019	0.9989	0.0019	0.0322
CWT-CNN [29]	0.9936	0.0020	0.9941	0.0018	0.9939	0.0019	0.9938	0.0019	5.4068
AlexNet [37]	0.9858	0.0044	0.9868	0.0039	0.9860	0.0040	0.9858	0.0043	57.0202
LTCN-IBLS	0.9999	0.0002	0.9999	0.0002	0.9999	0.0002	0.9999	0.0002	0.0165

**Table 11 sensors-23-05642-t011:** Average results of the ablation experiments for case 2 under noisy conditions.

Model	Accuracy		MP		MR		MF	
	MV	SD	MV	SD	MV	SD	MV	SD
OLTCN-IBLS	0.9036	0.0075	0.9227	0.0039	0.9032	0.0075	0.9019	0.0083
TLTCN-IBLS	0.9326	0.0041	0.9377	0.0036	0.9326	0.0041	0.9311	0.0043
OLTCN-TLTCN	0.9392	0.0014	0.9499	0.0012	0.9380	0.0014	0.9325	0.0017
LTCN-IBLS	0.9753	0.0015	0.9752	0.0014	0.9753	0.0015	0.9752	0.0014

**Table 12 sensors-23-05642-t012:** Average diagnosis results of different models for case 2 under noisy conditions.

Model	Accuracy		MP		MR		MF		Number of Trainable Parameters (M)
	MV	SD	MV	SD	MV	SD	MV	SD	
R-O-IBLS	0.9630	0.0033	0.9652	0.0030	0.9630	0.0033	0.9632	0.0033	0.0294
WDCNN [36]	0.9686	0.0039	0.9686	0.0039	0.9707	0.0033	0.9688	0.0038	0.0174
1-D DCNN [9]	0.9691	0.0020	0.9721	0.0016	0.9697	0.0022	0.9708	0.0005	0.0661
Deep ResNet [10]	0.9658	0.0028	0.9672	0.0025	0.9653	0.0025	0.9655	0.0026	0.0322
CWT-CNN [29]	0.9601	0.0015	0.9623	0.0020	0.9599	0.0016	0.9597	0.0015	5.4068
AlexNet [37]	0.9600	0.0053	0.9630	0.0042	0.9598	0.0052	0.9592	0.0051	57.0202
LTCN-IBLS	0.9753	0.0015	0.9752	0.0014	0.9753	0.0015	0.9752	0.0014	0.0165

**Table 13 sensors-23-05642-t013:** Details of the five rolling components.

Component	Inner Diameter (mm)	Fault Description
Bearing 1	50	Inner race defect
Bearing 2	50	Outer race defect
Bearing 3	50	Missing ball
Gear 1/Gear 2	55/50	Missing tooth

**Table 14 sensors-23-05642-t014:** Parameters of the main components of the experimental platform.

Name	Model	Parameter
MCACU	C8-2000	Rated power: 11 kW, maximum speed: 2970 rpm
Torque sensor	CYT-302	Rated torque: 100 Nm, speed range: 0–3000 rpm
Motor	YP-50-112	Rated voltage: 380 V, rated power: 11 kW
Velocity sensor	VE101-2D	Sensitivity: 4 mV/mm/sec

**Table 15 sensors-23-05642-t015:** Dataset details of case 3.

Label	Fault Description	Training Samples/Test Samples
0	Healthy	1400/600
1	Single fault (Gear 1)	1400/600
2	Compound fault (Gear 1 and Bearing 1)	1400/600
3	Compound fault (Gear 1 and Bearing 2)	1400/600
4	Compound fault (Gear 1 and Bearing 3)	1400/600
5	Compound fault (Gear 1 and Gear 2)	1400/600

**Table 16 sensors-23-05642-t016:** Average results of the ablation experiments for case 3.

Model	Accuracy		MP		MR		MF	
	MV	SD	MV	SD	MV	SD	MV	SD
OLTCN-IBLS	0.9394	0.0039	0.9406	0.0041	0.9397	0.0039	0.9390	0.0040
TLTCN-IBLS	0.9444	0.0030	0.9470	0.0030	0.9444	0.0030	0.9441	0.0031
OLTCN-TLTCN	0.9726	0.0023	0.9732	0.0023	0.9726	0.0023	0.9725	0.0023
LTCN-IBLS	0.9800	0.0022	0.9806	0.0020	0.9800	0.0022	0.9797	0.0022

**Table 17 sensors-23-05642-t017:** Average diagnosis results of the different models for case 3.

Model	Accuracy		MP		MR		MF		Number of Trainable Parameters (M)
	MV	SD	MV	SD	MV	SD	MV	SD	
R-O-IBLS	0.9763	0.0031	0.9785	0.0022	0.9770	0.0033	0.9769	0.0029	0.0296
WDCNN [36]	0.9766	0.0027	0.9773	0.0026	0.9766	0.0028	0.9764	0.0028	0.0113
1-D DCNN [9]	0.9691	0.0058	0.9709	0.0054	0.9692	0.0058	0.9691	0.0058	0.0225
Deep ResNet [10]	0.9740	0.0046	0.9755	0.0042	0.9743	0.0045	0.9740	0.0046	0.0104
CWT-CNN [29]	0.9396	0.0086	0.9467	0.0060	0.9396	0.0086	0.9386	0.0091	5.4069
AlexNet [37]	0.9638	0.0031	0.9658	0.0027	0.9639	0.0030	0.9636	0.0031	57.0284
LTCN-IBLS	0.9800	0.0022	0.9806	0.0020	0.9800	0.0022	0.9797	0.0022	0.0101

**Table 18 sensors-23-05642-t018:** Average results of the ablation experiments for case 3 under noisy conditions.

Model	Accuracy		MP		MR		MF	
	MV	SD	MV	SD	MV	SD	MV	SD
OLTCN-IBLS	0.8098	0.0045	0.8083	0.0049	0.8094	0.0046	0.8058	0.0050
TLTCN-IBLS	0.8068	0.0019	0.8128	0.0069	0.8061	0.0017	0.7845	0.0022
OLTCN-TLTCN	0.8910	0.0028	0.9123	0.0044	0.8920	0.0031	0.8826	0.0028
LTCN-IBLS	0.9158	0.0033	0.9235	0.0068	0.9158	0.0036	0.9148	0.0029

**Table 19 sensors-23-05642-t019:** Average diagnosis results of the different models for case 3 under noisy conditions.

Model	Accuracy		MP		MR		MF		Number of Trainable Parameters (M)
	MV	SD	MV	SD	MV	SD	MV	SD	
R-O-IBLS	0.8881	0.0035	0.8876	0.0034	0.8921	0.0045	0.8858	0.0040	0.0296
WDCNN [36]	0.9024	0.0045	0.9136	0.0035	0.9021	0.0047	0.8995	0.0056	0.0113
1-D DCNN [9]	0.9047	0.0036	0.9069	0.0040	0.9045	0.0036	0.9036	0.0035	0.0225
Deep ResNet [10]	0.8986	0.0050	0.8955	0.0024	0.8971	0.0028	0.8952	0.0026	0.0104
CWT-CNN [29]	0.8605	0.0076	0.8687	0.0053	0.8605	0.0074	0.8568	0.0100	5.4069
AlexNet [37]	0.8605	0.0049	0.8690	0.0091	0.8606	0.0052	0.8574	0.0055	57.0284
LTCN-IBLS	0.9158	0.0033	0.9235	0.0068	0.9158	0.0036	0.9148	0.0029	0.0101

**Table 20 sensors-23-05642-t020:** Comparison of CTTP for CWT-CNN, AlexNet, and LTCN-IBLS.

Model	CWRU Dataset	IMS Dataset	Gear and Bearing Dataset
CWT-CNN [29]	362.50 s	235.12 s	309.29 s
AlexNet [37]	1052.03 s	503.88 s	851.17 s
LTCN-IBLS	301.96 s	227.50 s	298.59 s

## Data Availability

Not applicable.

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
