# Peer review of "Fault Diagnosis of Rotating Machinery: A Highly Efficient and Lightweight Framework Based on a Temporal Convolutional Network and Broad Learning System"

_sensors, 2023, doi:10.3390/s23125642_

Round 1
Reviewer 1 Report
The paper discusses the problems of fault detection of rotational machinery using data-driven approach. The analysis is based on a well established data set. The authors present a so-called two-stage approach of making fault detection.
There are however some concerns regarding the presented results.
- The methodology is based on the models discussed in references [22] and [14]. Therefore the contributions of the manuscript should be properly discussed.
- The CRWU data set used in this paper has been extensively analysied using various methods. The authors made comparison only with other data driven methods. As seen in Table 4, the majority of those methods have a substantial number of parameters ranging from thousands to millions. However there are quite simple signal processing approaches that provide almost the same performance. I would suggest to check those results too.
- When discussing wavelet transform you must provide all the parameters, such as the mother wavelet, the selected number of voices per decade etc.
- Having done wavelet transform, using the resulting complete CWT map may be a major overkill. Since your machinery operates under constant condition it can be easily determined which bands are important. Simple wavelet based entropy denoising would've done the job.
- L202: Where is the value of 224 comes from?
- L212 the pytorch resize does linear interpolation. This to some extend renders the purpose of the CWT. Having fixed 224 points with spatio-temporal resolution can be achieved with simple SFFT.
- Concerning the above point the rationale behind equation (13) should be properly explained.
-Section 3 as complete is a little cumbersome. Since the main architectural blocks are from [14] and [22], I would suggest to describe the reasons why did the authors chose such an architecture. Written as such it seem like certain parts are just cobbled together without particular rationale. Since this is the main contribution of the paper, this section should be the most detailed one.
- Section 4 in global is rather non-informative. As I already pointed out, these datasets have been used in numerous research articles. Even with some simple signal processing approaches, such as spectral kurtosis with envelope analysis for bearings and side bands energy distribution of the gear-mesh frequency for gears can lead to a particularly precise information. I would suggest to check which are the best results overall on these datasets and compare with those. It does not have to be that the comparison should be done only with data-driven approaches.
- For example the photos shown in Figure 6 represent such a severe cases of pitting that even simple analysis of the vibration signals yields a satisfactory results. The main question is, what is so particular with the cases that are not properly classified?
- As written in L529, the network architecture should be described and that should be the focus of the paper.
- One minor note, I would remove working like novel and new. The paper will be here for quite a long time, and it won't be neither novel nor new in couple of years. So, instead of writing: 'We are proposing novel model...' just write we are proposing a model...
Author Response
Author response:
Concern #1:
The methodology is based on the models discussed in references [22] and [14]. Therefore the contributions of the manuscript should be properly discussed.
Answer:
Dear reviewer, thanks for your suggestions for discussing the two references. In the revised manuscript, the references [22] and [14] are changed to [31] and [17], respectively.
For the temporal convolutional network (TCN) proposed in [31], we had discussed its principle and structure in subsection 2.2 (Line 167-Line 176, revised manuscript). According to your suggestions, we further discussed and supplemented the contributions about TCN in Line 176-Line 181 of revised manuscript with track changes. The supplemented contents are
“TCNs have advantages over recurrent neural networks (RNNs) in concurrent data processing. The performance of TCNs has exceeded RNNs in many fields such as audio synthesis and machine translation [31]. Therefore, TCNs are contributed to researchers to get rid of the dependence of the RNNs in the tasks of sequence modeling and tem-poral features extraction, which is beneficial to the diversity of solutions.”
For the broad learning system with incremental learning of additional enhancement nodes proposed in [17], we had discussed its principle and structure in subsection 2.3 (Line 185-Line 206, revised manuscript). The contribution of reference [17] was descripted in Line 185-Line 188, that is, “Traditional DL algorithms typically suffer from a complex model structure, caus-ing time-consuming training. Chen and Liu [17] used a random vector function-al-link neural network and proposed the BLS to overcome the dependence on the deep structure of the DL algorithms.” In order to meet better your requirements, we further supplemented the discussion about reference [17] in Line 206-Line 210 of revised manuscript with track changes. The supplemented contents are
“The BLS has been widely used in classification and regression tasks due to its ad-vantages of fast training speed and high accuracy. Besides, the BLS enables researchers to model without limiting themselves to deep network structure. However, the accu-racy of BLS largely depends on the number of nodes, resulting in redundant nodes and trainable parameters [34].”
We hope such supplements can meet your requirements. Thank you.
Concern #2:
The CRWU data set used in this paper has been extensively analysied using various methods. The authors made comparison only with other data driven methods. As seen in Table 4, the majority of those methods have a substantial number of parameters ranging from thousands to millions. However there are quite simple signal processing approaches that provide almost the same performance. I would suggest to check those results too.
Answer:
Dear reviewer, thanks for your suggestions about signal processing approaches.
As you point out, the excessive number of parameters has always been a concern that hinders the development of dada-based models. As we introduced in Line 89-Line 91 of revised manuscript, “Another concern in fault diagnosis is the complexity (number of trainable param-eters) of models. Many trainable parameters require high computing power and reduce the practicality of the models.” Therefore, the lightweight of the models is one of the contributions of our manuscript. As shown in Table 5, Table 10 and Table 17 of revised manuscript, our proposed model has the lowest number of parameters, even lower than compared models (such as WDCNN, 1-D DCNN and Deep ResNet) with one-dimensional (1-D) input.
Signal processing-based fault diagnosis (SPFD) is classified as the traditional intelligent diagnosis method, which could be divided into two steps: (1) perform signal analysis of the monitoring data and manually select features; (2) assess faults artificially based on the signal analysis results or classify the faults with a classification algorithm [1-2]. However, there are some researches [3-6] show that the effectiveness of the traditional fault diagnosis method relies largely on the experience of the maintenance personnel, leading to variable diagnosis results or misdiagnosis, which is not conducive to industrial application. Besides, the non-stationarity of vibration signals and the nonlinear characteristics of rotating machinery in different scenarios could interfere with the extraction and identification of fault features [7]. Thus, we devote ourselves to research more intelligent methods to understand the relationship between the monitoring data and the equipment conditions to reduce the dependence on expert experience and difficulty of fault diagnosis.
Benefit by the development of big data and breakthrough in hardware technology, data-based models such as deep learning have developed rapidly and been widely used in the field of fault diagnosis [8]. As we introduced in Line 56-Line 60 of revised manuscript, “Deep learning-based fault diagnosis (DLFD), which can automatically extract features and establish the relationship between monitoring data and fault modes [8], has been increasingly applied to fault diagnosis research since it reduces the reliance on expert knowledge.” Therefore, our research in this manuscript focuses on the field of DLFD. However, this does not mean that signal processing-based fault diagnosis is worthless. We acknowledge that the signal processing-based fault diagnosis is also one of the research focuses in the field of fault diagnosis, and hope to explore this method with you in future research.
In order to meet your requirements, we supplemented the contents about the shortcomings of signal processing-based fault diagnosis in Line 43-Line 54 of revised manuscript with track changes. The supplemented contents are
“The development of artificial intelligence technology has attracted widespread attention for use in fault diagnosis based on monitoring data collected from sensors. Traditional intelligent fault diagnosis consists of two steps: (1) signal analysis of the monitoring data and manual features selection; (2) artificial fault assessment based on the signal analysis results and/or classification algorithms [2, 5]. However, the effectiveness of the traditional fault diagnosis method relies largely on the experience of the maintenance personnel, leading to variable diagnosis results [6]. Besides, the non-stationarity of vibration signals and the nonlinear characteristics of rotating machinery in different scenarios can interfere with the extraction and identification of fault features [7]. Thus, it is necessary to research more intelligent methods to under-stand the relationship between the monitoring data and the equipment conditions to reduce the dependence of fault diagnosis on expert experience.”
We hope such supplements can meet your requirements. Thank you.
Reference
[1] Han, T.; Ma, R.Y.; Zheng, J.G. Combination bidirectional long short-term memory and capsule network for rotating machinery fault diagnosis. Measurement. 2021, 176, 109208. DOI: 10.1016/j.measurement.2021.109208.
[2] Zhang, R.; Peng, Z.; Wu, L. F.; Yao, B. B.; Guan, Y. Fault Diagnosis from Raw Sensor Data Using Deep Neural Networks Considering Temporal Coherence. Sensors. 2017, 17, 549. doi:10.3390/s17030549.
[3] Peng, B. S.; Xia, H.; Lv, X. Z.; Annor-Nyarko, M.; Zhu, S. M.; Liu, Y. K.; Zhang, J. Y. An intelligent fault diagnosis method for rotating machinery based on data fusion and deep residual neural network. Appl. Intell. 2021. DOI: 10.1007/s10489-021-02555-4.
[4] Shao, S. Y.; McAleer, S.; Yan, R. Q.; Baldi, P. Highly Accurate Machine Fault Diagnosis Using Deep Transfer Learning. IEEE Trans. Ind. Informat. 2019, 15, 2446-2455. DOI: 10.1109/TII.2018.2864759.
[5] Fu, Y.; Cao, H.R.; Chen, X.F.; Adaptive Broad Learning System for High-Efficiency Fault Diagnosis of Rotating Machinery. IEEE Trans. Instrum. Meas. 2021, 70, 3519011. DOI: 10.1109/TIM.2021.3085940.
[6] Zhang, W.; Li, X.; Ding, Q. Deep residual learning-based fault diagnosis method for rotating machinery. ISA Transactions. 2019, 95, 295-305. DOI: 10.1016/j.isatra.2018.12.025.
[7] Liang, P. F.; Deng, C.; Wu, J.; Li, G. Q.; Yang, Z. X.; Wang, Y. H. Intelligent Fault Diagnosis via Semisupervised Generative Adversarial Nets and Wavelet Transform. IEEE Trans. Instrum. Meas. 2020, 69, 4659-4671. DOI: 10.1109/TIM.2019.2956613.
[8] Jiao, J. Y.; Zhao, M.; Lin, J.; Liang, K. X. A comprehensive review on convolutional neural network in machine fault diagnosis. Neurocomputing. 2020, 417, 36-63. DOI: 10.1016/j.neucom.2020.07.088.
[9] Saufi, S.R.; Ahmad, Z.A.B.; Leong, M.S.; Lim, M.H. Gearbox Fault Diagnosis Using a Deep Learning Model With Limited Data Sample. IEEE Trans. Ins. Informat. 2020, 16, 6263-6271. DOI: 10.1109/TII.2020.2967822.
Concern #3:
When discussing wavelet transform you must provide all the parameters, such as the mother wavelet, the selected number of voices per decade etc.
Answer:
Dear reviewer, thanks for your suggestions about the wavelet transform.
According to your suggestions, we supplemented the parameters of wavelet transform in Line 261-Line 263 of revised manuscript with track changes. The supplemented contents are
“The sampling period of the frequency is the same as that of the vibration signal. The Morlet wavelet is used in our work because its shape is similar to the pulse signal of mechanical faults, facilitating fault diagnosis [4].”
We hope such supplements can meet your requirements. Thank you.
Concern #4:
Having done wavelet transform, using the resulting complete CWT map may be a major overkill. Since your machinery operates under constant condition it can be easily determined which bands are important. Simple wavelet based entropy denoising would've done the job.
Answer:
Dear reviewer, We apologize for not giving a good reason for using the wavelet transform to process the data.
As shown in the Line 60-Line 85 of revised manuscript with track changes, we illustrated the respective shortcomings of two subfields ((1) fault diagnosis based on raw monitoring data and (2) fault diagnosis based on time-frequency domain transformation of monitoring data.) of deep learning-based fault diagnosis.
For the subfield (1), there has a review literature [1] shows that many mechanical faults will generate diversified specific dynamic response signals. Besides, the raw vibration signal is complicated and non-stationary because of the structure and complexity of the equipment, resulting in an inability to identify the fault features of the rotating machinery accurately in many applications. For the subfield (2), the time-frequency analysis of the raw data could corrupt the structure of the vibration signals and reduce the original fault features, which would lead to the one-sidedness of the fault feature representation and thus affect the accuracy of fault identification [2-3]. Therefore, one of the goals of our manuscript is to propose a method to compensate for the shortcomings of the two subfields by extracting comprehensive fault features that complements the fault information contained in the time-domain features and time-frequency domain features.
Therefore, in our manuscript, the role of the wavelet transform is to show the frequency change of the vibration signal over time, which served as the input of our proposed 2-D LTCN backbone. In order to meet your requirements, we supplemented the reason why we used wavelet transform to process signal rather than other algorithms such as FT and STFT in Line 140-Line 153 of revised manuscript with track changes. The supplemented contents are
“Suitable data can significantly improve the performance of DLFD models. With the development of sensor technology, many types of sensors, such as accelerometers and built-in encoders, have been used for mechanical condition monitoring [21]. The temporal signal collected by a sensor during working conditions is complex and non-stationary, complicating the identification of the fault features [11, 22]. Therefore, signal-processing algorithms, such as Fourier transform (FT), short-time Fourier trans-form (STFT), and wavelet transform (WT), are useful for fault diagnosis. FT is a widely used signal-processing algorithm but has several disadvantages [23]. It cannot display the local frequency-domain features of the signal, and the corresponding time-domain information prevents the accurate processing of non-stationary signals. Although STFT can process non-stationary signals, it cannot change the time-frequency resolution during signal processing [24]. The WT has more basic functions and can perform mul-ti-resolution signal analysis; thus, it is a preferred algorithm for fault diagnosis of ro-tating machinery [25-26]. Therefore, the WT is employed in our work.”
We hope our explanations and supplements can meet your requirements. Thank you.
Reference
[1] Chen, J.L.; Li, Z.P.; Pan, J.; Chen G.G.; Zi, Y.Y.; Yuan, J.; Chen, B.Q.; He, Z.J. Wavelet transform based on inner product in fault diagnosis of rotating machinery: A review. Mech. Syst. Signal Process. 2016, 70-71, 1-35. DOI: 10.1016/j.ymssp.2015.08.023.
[2] Meng, Z.; Guo X.L.; Pan, Z.Z.; Sun, D.Y.; Liu, S. Data Segmentation and Augmentation Methods Based on Raw Data Using Deep Neural Networks Approach for Rotating Machinery Fault Diagnosis. IEEE Access. 2019, 7, 79510-79522. DOI: 10.1109/ACCESS.2019.2923417.
[3] Li, G.Q.; Wu, J.; Deng, C.; Chen, Z.Y.; Shao, X.Y. Convolutional Neural Network-Based Bayesian Gaussian Mixture for In-telligent Fault Diagnosis of Rotating Machinery. IEEE Trans. Instrum. Meas. 2021, 70, 3517410. DOI: 10.1109/TIM.2021.3080402.
Concern #5:
L202: Where is the value of 224 comes from?.
Answer:
Dear reviewer, we are sorry for the concern we brought to you.
For the models with two-dimensional input, the input size of 224×224 is an empirical value, and there is no exact theoretical support at present. In order to make our paper more rigorous, we removed the compression of the time-frequency image to the size of 224×224 and reverified the performance of our proposed model.
We hope such modifications can meet your requirements. Thank you.
Concern #6:
L212 the pytorch resize does linear interpolation. This to some extend renders the purpose of the CWT. Having fixed 224 points with spatio-temporal resolution can be achieved with simple SFFT.
Answer:
Dear reviewer, thanks for your suggestions about the signal preprocessing methods.
According the answer of Concern #5, we removed the operation of compressing the time-frequency image to the size of 224×224, which also can avoid incomplete information caused by data compression.
Thank you again for your suggestions.
Concern #7:
Concerning the above point the rationale behind equation (13) should be properly explained.
Answer:
Dear reviewer, thanks for your suggestion for further explaining the equation (13).
According to the answer of concern #5 and concern #6, we removed the operation to compress the time-frequency image to the size of 224×224. Therefore, the equation (13) of original manuscript was removed. The number of data points contained in each sample can be determine by the equation (12) of the revised manuscript with track changes, that is,
where is the number of data points contained in each sample, is the number of sampling points per rotation of the equipment, is the sampling frequency, and is the rotation speed of the equipment.
Besides, the reason why determine N based on the equation (12) was introduced in Line 248-Line 253 of the revised manuscript with track changes. The reason is
“We consider the following two factors to determine the value of N. First, each sample contains enough data points to represent the fault features of a full rotation cycle of the equipment. Second, we minimize N to satisfy the first two factors and reduce the computational cost.”
We hope such modifications can meet your requirements. Thank you.
Concern #8:
Section 3 as complete is a little cumbersome. Since the main architectural blocks are from [14] and [22], I would suggest to describe the reasons why did the authors chose such an architecture. Written as such it seem like certain parts are just cobbled together without particular rationale. Since this is the main contribution of the paper, this section should be the most detailed one.
Answer:
Dear reviewer, thanks for your suggestions for the Section 3.
According to your suggestions, we modified the Section 3 and supplemented the reasons why we proposed such an architecture in Line 236-Line 244 and Line 308-Line 311 of revised manuscript with track changes. The supplemented contents about the reasons are
“In traditional fault diagnosis models, the extracted features must be input into the fully connected classifier for fault identification. However, stacking multiple dense layers is generally required to meet diagnostic requirements and improve accuracy, resulting in excessive trainable parameters for the classifier. Therefore, we replace the dense layers with the IBLS to achieve a lightweight classifier. In addition, we propose a feature ex-traction stage (stage 1) with a lightweight network structure to perform adaptive representation learning of data before identifying the fault classes to minimize the redundancy of the IBLS nodes and trainable parameters and achieve high accuracy.”
“After completing the representation learning of data in stage 1, we input the extracted fault features into the fault diagnosis stage (stage 2) and adopt the IBLS to replace the fully connected classifier to improve the ability to establish complex nonlinear relationships between features and fault classes.”
We hope such modifications and supplements can meet your requirements. Thank you.
Concern #9:
Section 4 in global is rather non-informative. As I already pointed out, these datasets have been used in numerous research articles. Even with some simple signal processing approaches, such as spectral kurtosis with envelope analysis for bearings and side bands energy distribution of the gear-mesh frequency for gears can lead to a particularly precise information. I would suggest to check which are the best results overall on these datasets and compare with those. It does not have to be that the comparison should be done only with data-driven approaches.
Answer:
Dear reviewer, thanks for your suggestions about the signal processing approaches.
The main research focus of signal processing-based fault diagnosis is to propose methods based on signal processing algorithms (such as spectral kurtosis) that can effectively separate features of different faults, which can further enrich fault information and realize fault diagnosis [1-3]. However, as introduced in the answer of concern #2, that is,
“there are some researches [4-7] show that the effectiveness of the traditional fault diagnosis method relies largely on the experience of the maintenance personnel, leading to variable diagnosis results or misdiagnosis, which is not conducive to industrial application. Besides, the non-stationarity of vibration signals and the nonlinear characteristics of rotating machinery in different scenarios could interfere with the extraction and identification of fault features [8]. Thus, we devote ourselves to research more intelligent methods to understand the relationship between the monitoring data and the equipment conditions to reduce the dependence on expert experience and difficulty of fault diagnosis.
Benefit by the development of big data and breakthrough in hardware technology, data-based models such as deep learning have developed rapidly and been widely used in the field of fault diagnosis [9]. As we introduced in Line 56-Line 60 of revised manuscript, “Deep learning-based fault diagnosis (DLFD), which can automatically extract features and establish the relationship between monitoring data and fault modes [10] has been increasingly applied to fault diagnosis research. They reduce the reliance on expert knowledge and have been increasingly applied to fault diagnosis research.” Therefore, our research in this manuscript focuses on the field of DLFD.”
Besides, the evaluation matrix to evaluate the performance of the data-based fault diagnosis methods are accuracy, precision, recall, etc., which is different from the methods of signal processing-based fault diagnosis. Therefore, it could be difficult to directly compare data-based methods to signal processing-based methods. We look forward to exploring hot research in signal processing-based fault diagnosis with you in the future, and thank you again for your suggestions on our manuscript.
We hope such explanations can address your concern. Thank you.
Reference
[1] Xiang, L.; Li, J.X.; Hu, A. J.; Li, Y. Separation and Extraction of Composite Fault Characteristics of Wind Turbine Bearing Based on SK⁃MOMEDA. Journal of Vibration,Measurement & Diagnosis. 2021, 41, 644-651. DOI: 10.16450/j.cnki.issn.1004⁃6801.2021.04.002.
[2] Xiang, J. W.; Zhong, Y. T.; Gao, H. F. Rolling element bearing fault detection using PPCA and spectral kurtosis. Measurement. 2015, 75, 180–191. DOI: 10.1016/j.measurement.2015.07.045z.
[3] Wang, T. Y.; Chu, F. L.; Han, Q. K.; Kong, Y. Compound faults detection in gearbox via meshing resonance and spectral kurtosis methods. Journal of Sound and Vibration. 2017, 392, 367–381. DOI: 10.1016/j.jsv.2016.12.041.
[4] Peng, B. S.; Xia, H.; Lv, X. Z.; Annor-Nyarko, M.; Zhu, S. M.; Liu, Y. K.; Zhang, J. Y. An intelligent fault diagnosis method for rotating machinery based on data fusion and deep residual neural network. Appl. Intell. 2021. DOI: 10.1007/s10489-021-02555-4.
[5] Shao, S. Y.; McAleer, S.; Yan, R. Q.; Baldi, P. Highly Accurate Machine Fault Diagnosis Using Deep Transfer Learning. IEEE Trans. Ind. Informat. 2019, 15, 2446-2455. DOI: 10.1109/TII.2018.2864759.
[6] Fu, Y.; Cao, H.R.; Chen, X.F.; Adaptive Broad Learning System for High-Efficiency Fault Diagnosis of Rotating Machinery. IEEE Trans. Instrum. Meas. 2021, 70, 3519011. DOI: 10.1109/TIM.2021.3085940.
[7] Zhang, W.; Li, X.; Ding, Q. Deep residual learning-based fault diagnosis method for rotating machinery. ISA Transactions. 2019, 95, 295-305. DOI: 10.1016/j.isatra.2018.12.025.
[8] Liang, P. F.; Deng, C.; Wu, J.; Li, G. Q.; Yang, Z. X.; Wang, Y. H. Intelligent Fault Diagnosis via Semisupervised Generative Adversarial Nets and Wavelet Transform. IEEE Trans. Instrum. Meas. 2020, 69, 4659-4671. DOI: 10.1109/TIM.2019.2956613.
[9] Jiao, J. Y.; Zhao, M.; Lin, J.; Liang, K. X. A comprehensive review on convolutional neural network in machine fault diagnosis. Neurocomputing. 2020, 417, 36-63. DOI: 10.1016/j.neucom.2020.07.088.
[10] Saufi, S.R.; Ahmad, Z.A.B.; Leong, M.S.; Lim, M.H. Gearbox Fault Diagnosis Using a Deep Learning Model With Limited Data Sample. IEEE Trans. Ins. Informat. 2020, 16, 6263-6271. DOI: 10.1109/TII.2020.2967822.
Concern #10:
For example the photos shown in Figure 6 represent such a severe cases of pitting that even simple analysis of the vibration signals yields a satisfactory results. The main question is, what is so particular with the cases that are not properly classified?
Answer:
Dear reviewer, we are sorry for the concern we brought to you. The Figure 6 of the original manuscript was changed to Figure 7 in the revised manuscript.
Figure 7 shows a picture of the fault components in case 3, which were used to form the six fault classes shown in Table 14 of revised manuscript. In order to better meet your requirements, we supplemented the confusion matrixes (show in Figure 9 and Figure 10 in our revised manuscript) of our proposed model in case 3 to clearly show the misclassified faults. We supplemented the description about the Figure 9 and Figure 10 in Line 534-Line 537 and Line 556-Line 559 of revised manuscript with track changes, respectively. The supplemented contents are
“The confusion matrix of LTCN-IBLS for the test dataset is shown in Figure 9. The samples of the healthy condition (label 0) and single fault (label 1) are all correctly classified. Some samples of compound faults are incorrectly classified, and 60 samples with label 2 are misclassified.”
“The confusion matrix of the LTCN-IBLS framework is presented in Figure 10. All samples of the healthy condition (label 0) are all correctly classified, and only one sample of the single fault is misclassified. However, there are many misclassification cases in the samples of the compound faults (especially label 2 and label 4).”
Therefore, we can summarize that the misclassified samples are mainly compound faults (label 2-label 5 shown in Table 14). It is more difficult to diagnose compound faults than single faults. We supplemented the conclusion in Line 646-Line 651 of revised manuscript with track changes. The supplemented contents are
“Figure 9 and Figure 10 indicate that it is more difficult to diagnose compound faults than single faults. Therefore, we will focus on more effective intelligent algorithms for improving the calculation speed of the models and diagnosing the compound faults of rotating machinery in future studies.”
We hope such explanations and supplements can addressed your concern. Thank you.
Concern #11:
As written in L529, the network architecture should be described and that should be the focus of the paper.
Answer:
Dear reviewer, thank you for your suggestions about our network architecture.
According to your suggestions, we modified the description about our proposed model in Line 618-Line 631 of revised manuscript with track changes. The whole conclusion is shown as follow and the description about architecture and principle of our proposed network framework are marked in bold.
“In this study, we proposed an intelligent framework (called LTCN-IBLS) consisting of a feature extraction stage and a fault identification stage for data representation learning and faults diagnosis. In the feature extraction stage, the 1-D LTCN and 2-D LTCN backbone were used to extract the time-dependent information, including the time-frequency features and temporal features. Information from the future input in the direction of time axis was removed, while the complete frequency edge and corner information was retained during the extraction of the time-frequency features. The time-frequency features were fused with the temporal features to obtain more in-depth and high-quality information on the fault features to improve the fault identification performance of the IBLS classifier in the fault identification stage. The IBLS classifier established an accurate mapping relationship between the fused features and the fault categories. It exhibited better nonlinear mapping ability than the fully connected layers. Ablation experiments demonstrated the rationality and contributions of the frame-work’s components. Under non-noisy conditions, the MVs of the accuracy, MP, MR, and MF of the LTCN-IBLS were up to 0.0560, 0.0426, 0.0560, and 0.0584 higher than those of the comparable models. Under noisy conditions, the MVs of the accuracy, MP, MR, and MF of the LTCN-IBLS were up to 0.1446, 0.1365, 0.1444, and 0.1492 higher than those of the comparable models. The LTCN-IBLS had the lowest number of trainable parameters (≤ 0.0165 M) among all models. The experimental results prove that the proposed framework possesses effectiveness, lightweight, and robustness for fault diagnosis.
This study provided insights and solutions for establishing lightweight neural network models to diagnose faults of rotating machinery, minimizing manual intervention. However, due to the large amount of computation in processing 2-D data, our proposed model is slower than many other models with 1-D input, although it has a low parameter number. Figure 9 and Figure 10 indicate that it is more difficult to diagnose compound faults than single faults. Therefore, we will focus on more effective intelligent algorithms for improving the calculation speed of the models and diagnosing the compound faults of rotating machinery in future studies.”
We hope the modifications and supplements can meet your requirements. Thank you.
Concern #12:
One minor note, I would remove working like novel and new. The paper will be here for quite a long time, and it won't be neither novel nor new in couple of years. So, instead of writing: 'We are proposing novel model...' just write we are proposing a model...
Answer:
Dear reviewer, thank you for your kindly note.
According to your note, we removed all the words such as novel and new. Thank you again for your time and professional suggestions about our manuscript.

Reviewer 2 Report
This article describes about a fault diagnosis of rotating machinery: A highly efficient and lightweight framework based on a TCN and broad learning system. In order to have your paper ready for publications, however, you should answer and revise the paper following comments:
1. Organize the advantages and disadvantages of the proposed method and other methods into a table and add it in the chapter 1 or 2.
2. Write a flowchart of the fault diagnosis procedure using propsed method and add it to the paper.
3. Add a table of comparing the time to run the fault diagnosis program for each algorithm in the Chapter 4.
4. Should supplement the conclusion based on the fault diagnosis results of the experiment.
Please check the English again!
Author Response
Author response:
Concern #1:
Organize the advantages and disadvantages of the proposed method and other methods into a table and add it in the chapter 1 or 2.
Answer:
Dear reviewer, thanks for your suggestions for summarizing the advantages and disadvantages of methods in our manuscript.
According to your suggestions, we added a table (Table 1) in the Section 1 of revised manuscript. In the Table 1, the main differences, advantages and disadvantages of different models are summarized. The descriptions about Table 1 were supplemented in Line 110-Line 113 of revised manuscript with track changes.
We hope such supplements can meet your requirements. Thank you.
Concern #2:
Write a flowchart of the fault diagnosis procedure using propsed method and add it to the paper.
Answer:
Dear reviewer, thanks for your suggestions about our proposed method.
According to your suggestions, we added a figure (Figure 4) to descript the flowchart of the fault diagnosis using our proposed method.
We hope such supplements can meet your requirements. Thank you.
Concern #3:
Add a table of comparing the time to run the fault diagnosis program for each algorithm in the Chapter 4.
Answer:
Dear reviewer, thanks for your suggestions about comparing the time between different models.
In the experiments of our manuscript, denote the dimension of the samples input to the models with one-dimensional (1-D) input is N, then the dimension of the samples input to the models with two-dimensional (2-D) input is N2. Therefore, the computational amount of the models with 2-D input is significantly more than that of the models with 1-D input, which makes it unfair to compare the running time of these two types of models. In the architecture of our proposed model, the input of the proposed 2-D LTCN backbone is 2-D time-frequency, which causes our proposed model to run slower that the models with only 1-D input. However, the main objectives of our manuscript are as follows. First, our proposed model has the highest evaluation metrics (accuracy, macro-precision, macro-recall and macro-F1 score) of fault diagnosis compared with other models. Second, the number of trainable parameters of the proposed method is lowest compared with other models. The experimental results on three datasets showed that our proposed LTCN-IBLS framework achieved the highest evaluation metrics and lowest number of trainable parameters compared with the other state-of-the-art fault diagnosis models.
In order to better meet your requirements, we performed a comparison of the convergence time of training process between models (CWT-CNN, AlexNet and LTCN-IBLS) with 2-D data processing capability. The results were shown in Table 20 of revised manuscript and the descriptions about Table 20 were supplemented in Line 610-Line 615 of revised manuscript with track changes.
Besides, running slower than models with 1-D input is analyzed as a shortcoming of our proposed model in Line 644-Line 646 of revised manuscript with track changes. We also supplemented that improving the proposed model running speed is one of our research focuses in the future in Line 648-Line 651 of revised manuscript with track changes.
We hope such modifications and supplements can meet your requirements. Thank you.
Concern #4:
Should supplement the conclusion based on the fault diagnosis results of the experiment.
Answer:
Dear reviewer, thank you for your suggestions for the conclusion.
According to your suggestions, in Line 634-Line 639 of revised manuscript with track changes, we modified and supplemented the most prominent performance of our proposed model based on the results of fault diagnosis. The supplemented contents are
“Under non-noisy conditions, the MVs of the accuracy, MP, MR, and MF of the LTCN-IBLS were up to 0.0560, 0.0426, 0.0560, and 0.0584 higher than those of the comparable models. Under noisy conditions, the MVs of the accuracy, MP, MR, and MF of the LTCN-IBLS were up to 0.1446, 0.1365, 0.1444, and 0.1492 higher than those of the comparable models. The LTCN-IBLS had the lowest number of trainable parameters (≤ 0.0165 M) among all models.”
We hope our explanations and supplements can meet your requirements. Thank you.

Reviewer 3 Report
In this paper was proposed a framework which consisting of a 1-D lightweight temporal convolutional network backbone, a 2-D backbone, and a BLS with incremental learning classifier to diagnose faults of rotating machinery.
This paper represents a significant contribution to machine diagnostics, but some minor additions are needed:
1. Revise the abstract by adding a summary of the main experimental results. Add one or two sentences in the abstract to show the vital results of the proposed approach.
2. The conclusion section has to show the impact and insights of this research work.
3. Need checking the English language and style.
Minor editing of English language required
Author Response
Author response:
Concern #1:
Revise the abstract by adding a summary of the main experimental results. Add one or two sentences in the abstract to show the vital results of the proposed approach.
Answer:
Dear reviewer, thanks for your suggestions for the abstract.
According to your suggestions, we modified and supplemented the most prominent performance of our proposed model based the experimental results in Line 25-Line 29 of revised manuscript with track changes. The supplemented contents are
“The results show that our framework provides the highest mean values of the evaluation metrics (accuracy ≥ 0.9158, MP ≥ 0.9235, MR ≥ 0.9158, and MF ≥ 0.9148) and the lowest number of trainable parameters (≤ 0.0165 Mage), indicating its high effectiveness and strong robustness for fault diagnosis.”
We hope such modifications and supplements can meet your requirements. Thank you.
Concern #2:
The conclusion section has to show the impact and insights of this research work.
Answer:
Dear reviewer, thanks for your suggestions for the conclusion.
According to your suggestion, we modified and supplemented the impact and insights of our work in Line 641-Line 642 of revised manuscript with track changes. The supplemented contents are
“This study provided insights and solutions for establishing lightweight neural network models to diagnose faults of rotating machinery, minimizing manual intervention.”
We hope such supplements can meet your requirements. Thank you.
Concern #3:
Need checking the English language and style.
Answer:
Dear reviewer, thanks for your suggestions about the English language and style of our manuscript.
According to your suggestions, we improved the grammar and language once again by a 3rd party service for language polishing.
Thank you again for your support and professional suggestions for our manuscript.

Round 2
Reviewer 2 Report
The comments and suggestions of the reviewer were well reflected and revised.